# Evaluation of InSAR Tropospheric Delay Correction Methods in a Low-Latitude Alpine Canyon Region

**Yanxi Zhao, Xiaoqing Zuo \*, Yongfa Li, Shipeng Guo, Jinwei Bu**  **and Qihang Yang**

School of Land Resources Engineering, Kunming University of Science and Technology, Kunming 650093, China
\* Correspondence: zxq@kust.edu.cn

**Abstract:** Tropospheric delay error must be reduced during interferometric synthetic aperture radar (InSAR) measurement. Depending on different geographical environments, an appropriate correction method should be selected to improve the accuracy of InSAR deformation monitoring. In this study, surface deformation monitoring was conducted in a high mountain gorge region in Yunnan Province, China, using Sentinel-1A images of ascending and descending tracks. The tropospheric delay in the InSAR interferogram was corrected using the Linear, Generic Atmospheric Correction Online Service for InSAR (GACOS) and ERA-5 meteorological reanalysis data (ERA5) methods. The correction effect was evaluated by combining phase standard deviation, semi-variance function, elevation correlation, and global navigation satellite system (GNSS) deformation monitoring results. The mean value of the phase standard deviation (Aver) of the linear correction interferogram and the threshold value (sill) of the semi-variogram were reduced by –20.98% and –41%, respectively, while the accuracy of the InSAR deformation points near the GNSS site was increased by 58%. The results showed that the three methods reduced the tropospheric delay error of InSAR deformation monitoring by different degrees in low-latitude mountains and valleys. Linear correction was the best at alleviating the tropospheric delay, followed by GACOS, while ERA5 had poor correction stability.

**Keywords:** SBAS-InSAR; tropospheric delay; deformation monitoring; Jinsha River basin



## 1. Introduction

Interferometric synthetic aperture radar (InSAR) is characterized by active imaging, cloud penetration, and all-day operation. It is an important observation method that measures large-scale surface deformation and provides technical support for monitoring Earth surface movement. Atmospheric delay error is a significant source of error in repeated orbit InSAR measurements. Satellite radar signals are affected by atmospheric delays in the propagation process of the Earth's atmosphere, and because real surface deformation signals are easily masked, their extraction is difficult [1]. As C-band SAR data were used in this study, the C-band ionospheric delay signal was small [2]; therefore, this paper only studied tropospheric delay.

Tropospheric delay is the phase delay caused by changes in tropospheric atmospheric parameters, such as pressure, temperature, and water vapor content; changes in water vapor content significantly affect the tropospheric delay [3]. Zebker et al. [4] found that for SIR-C/X-SAR, a 20% relative humidity variation in space–time led to an error of approximately 10 cm in the InSAR deformation monitoring results. Tropospheric delay noise greatly limits the signal range of spaceborne InSAR measurements and affects the confidence of researchers in inferring the nature of Earth's motion [5]. For repeated orbital interferometry, because the atmospheric conditions of two SAR imaging periods are different during satellite transit and space-borne, InSAR is affected by the tropospheric delay and the deformation measurement accuracy is greatly reduced.

Atmospheric delay is the primary limitation of InSAR technology [6], in which tropospheric delay is an important error that must be corrected in space-borne InSAR deformation measurements. Many scholars have conducted research on tropospheric delay error in

space-borne InSAR deformation measurements. Tropospheric delay can be divided into vertical stratification delay and turbulent mixing delay. The vertical stratification delay is strongly correlated with the terrain, and the turbulence mixing delay exhibits strong randomness and is independent of the terrain. At present, common correction methods for tropospheric delay include the following: (1) The empirical model correction method. Bekaert et al. [7] considered the spatial variability of the atmosphere and proposed a power law model for tropospheric correction. After correction, the correlation between the global navigation satellite system (GNSS)- and InSAR-estimated slow slip surface deformation was improved. (2) Correction method for estimating tropospheric delay in global positioning and navigation systems. Song et al. [8] utilized a double difference algorithm to correct the InSAR atmospheric delay effect pixel-by-pixel and used the atmospheric delay obtained from GNSS stations to calculate the delay of unknown pixels and verify the effectiveness of the method. (3) High-resolution numerical atmospheric model correction method. Zhang et al. [9] selected nine regions in China, covering interferograms of different climate types, seasons, and topography. The nonlinear correction effect of ERA-Interim reanalysis (ERA-I) and ERA-5 meteorological reanalysis data (ERA5) on tropospheric delay in the nine regions were analyzed; the results showed that the two reanalysis methods were best in winter and worst in summer in the northern hemisphere of China, and the correction effects varied with climate type. (4) Generic Atmospheric Correction Online Service for InSAR (GACOS). Xiao et al., used the GACOS product to correct the InSAR tropospheric delay of two typical terrain-like areas in eastern China and evaluated the correction effect, verifying that it could alleviate tropospheric delay [10]. (5) Machine learning correction method. Ghosh et al. [11] proposed a method based on a generative adversarial network (GAN) to mitigate the phase delay caused by the troposphere and realize a noise-to-noise model. Compared with the original interferogram, this method reduced the root mean square (RMS) of the phase value of the interferogram by 64%. Moreover, the correction effect was better than that of ERA-interim meteorological data correction.

These tropospheric delay correction methods are restricted by varied factors, and their correction effects are different. For example, the empirical model correction method cannot capture the turbulent mixing delay in the troposphere, while the correction method for estimating the tropospheric delay using a GNSS navigation system is limited by the distribution density of GNSS sites in the region of interest. Generally, the spatial distribution of GNSS sites is sparse, and the spatial resolution of tropospheric delay correction is low, despite the lack of GNSS sites in some regions of interest; therefore, this method cannot be used. The estimation results of the turbulence delay component based on the correction method using a high-resolution atmospheric numerical model are poor [12] and the spatial resolution of the meteorological data is low. Therefore, the tropospheric delay may not be reasonably reduced at different times and regions. The zenith tropospheric delay product (ZTD) provided by the GACOS uses meteorological data at 6 h intervals with a low time resolution. The machine learning correction method requires time to train the model, greatly reducing the efficiency of deformation monitoring data processing and occupying considerable computer memory [11].

At present, despite the many methods for InSAR tropospheric correction, none can be adopted as a universal correction method for application in different areas. This study aimed to determine a reasonable tropospheric delay correction method that considers different geographical environments, to improve the accuracy of InSAR deformation monitoring. The authors considered that the study area is located in a low-latitude alpine canyon area with large topographic relief, and no GNSS monitoring point with uniform coverage is found within it. Therefore, the empirical model related to topographic relief (Linear correction method), the GACOS method with the highest spatial resolution, and the high-resolution atmospheric numerical model correction method (ERA5 dataset with the highest temporal resolution) were used for tropospheric delay correction, and the suitability of the three methods in this low-latitude alpine canyon region was analyzed and evaluated. Through comparative analysis, it was found that the Linear method was the

best method to reduce the tropospheric delay error in these low-latitude mountains and valleys, maximizing the accuracy of InSAR deformation monitoring.

## 2. Materials and Methods

### 2.1. InSAR Tropospheric Delay

Changes in pressure, temperature, and relative humidity in the lower troposphere produce a tropospheric effect that generates a 15–20 cm signal in interferograms, which is usually much larger than the structural signal of interest [2]. The tropospheric delay phase caused by the atmosphere can be defined by the atmospheric refractive index $N$, which consists of the hydrostatic refraction component $N_{hydro}$ and wet refraction component $N_{wet}$:

$$N = (k_1 \frac{P}{T})_{hydro} + (k_2' \frac{e}{T} + k_3 \frac{e}{T^2})_{wet} = N_{hydro} + N_{wet} \tag{1}$$

where $P$ represents the total atmospheric pressure (hPa), $T$ represents the temperature (Kelvin), and $e$ represents the partial pressure of water vapor (hPa). $k_1$, $k_2'$, and $k_3$ are regarded as empirical constants, and their general values are $k_1 = 77.6$ K·hPa$^{-1}$, $k_2' = 23.3$ K·hPa$^{-1}$, and $k_3 = 3.75 \times 10^5$ K$^2$·hPa$^{-1}$ [13].

When the bidirectional tropospheric delay phase $\phi_{tropo}$ is at a specific altitude, $h = h_1$, the integral value of the refractive index between $h_1$ and $h_{top}$ at the top of the troposphere along the radar line of sight is given by Equation (2):

$$\phi tropo = \frac{-4\pi}{\lambda} \cdot \frac{10^{-6}}{cos\theta} \int_{h_1}^{h_{top}} \left( N_{hydro} + N_{wet} \right) dh \tag{2}$$

where $\lambda$ is the radar wavelength, and $\theta$ is the incidence angle. For heavy-orbit InSAR interferometry technology, the tropospheric conditions at the two imaging times are different, and the tropospheric delay $\Delta\phi_{tropo}$ between the reference and auxiliary images also varies. Therefore, the tropospheric delay phase in the interferogram is the difference in the tropospheric delay at the imaging point between the reference and auxiliary images. This delay phase depends on the relative change in atmospheric conditions at the time of the two images, rather than the value of the tropospheric delay generated at the time of a single image. $\phi_{tropo}^{ref}$ is the tropospheric delay phase value at the time of reference image imaging, and $\phi_{tropo}^{slv}$ is the tropospheric delay phase value at the time of auxiliary image imaging.

$$\Delta\phi_{tropo} = \phi_{tropo}^{slv} - \phi_{tropo}^{ref} \tag{3}$$

#### 2.1.1. Linear Correction Method

The temperature and pressure of dry air in the atmosphere are mainly stratified vertically, resulting in a large phase delay in radar signals that only changes with altitude [14], and a vertical stratified delay strongly related to terrain is introduced. In contrast, water vapor in the air varies vertically and laterally over short distances [14], introducing local random turbulent mixing delays, independently of topography. Vertical stratification in the tropospheric delay is static within a certain region and time [15]. If the interferogram is only affected by the vertical stratification delay, assuming that its tropospheric delay has a linear relationship with the terrain, then the interferogram tropospheric delay phase $\Delta\phi_{tropo\_linear}$ between the reference image and the auxiliary image estimated by the linear correction method can be expressed as

$$\Delta\phi_{tropo\_linear} = k_{\Delta\phi} h + \Delta\phi_0 \tag{4}$$

where $k_{\Delta\phi}$ is the coefficient of the tropospheric phase related to the topography of the interferogram, $h$ is the elevation, and $\Delta\phi_0$ represents the constant displacement applied to the entire interferogram, which can be ignored [7].

The empirical model (linear) correction method is applicable to most weather conditions except for particular cases, such as inverted or non-monotonic convective stratification [15]. At the same time, this method does not require the use of external data and can be used when external meteorological or GNSS data are lacking.

### 2.1.2. Generic Atmospheric Correction Online Service for InSAR (GACOS) Correction Method

GACOS products have global coverage and are free to use [16]. GACOS extends the iterative tropospheric decomposition (ITD) model [17], integrating $0.125° \times 0.125°$ horizontal resolution, 137 levels of vertical resolution, and high-resolution European Centre for Medium-Range Weather Forecasts (ECMWF) numerical weather models at 6 h intervals, and is combined with GNSS time-continuous high-precision point-by-point ZTD measurement data to generate a tropospheric correction map. The vertical and turbulent mixing delays can be estimated, and the ZTD includes stratification and turbulent mixing delays, as shown in Equation (5) [18]:

$$ZTD_k = S(h_k) + T(\mathbf{x_k}) + \varepsilon k \tag{5}$$

where $ZTD_k$ is the integration ZTD delay of the GNSS station and the ECMWF at position $k$, $T$ represents the turbulent component, and $\mathbf{x_k}$ is the station coordinate vector in the local geocentric coordinate system. $S$ represents the stratified component associated with height $h$, and $\varepsilon$ represents the remaining unmodeled residuals, including unmodeled stratification and turbulence signals. The layered-component model is as follows:

$$S_i = L_0 e^{-\beta h} => \begin{cases} S_m^G = L_0 e^{-\beta h_m} \\ S_n^E = L_0 e^{-\beta h_n} \end{cases}, P_i = \begin{bmatrix} P_G & 0 \\ 0 & P_E \end{bmatrix} \tag{6}$$

where $\beta$ is the exponential function coefficient of the stratified delay $S$, $L_0$ is the stratified component delay at sea level, $h$ is the height, $S_m^G$ represents the ZTD delay of GNSS at position $m$, $S_n^E$ represents the ZTD delay of ECMWF at position $n$, $P_i$ is the weight matrix, and the mass of tropospheric delay is defined in terms of GNSS and ECMWF reference positions.

The turbulence mixing delay part of the model is expressed as

$$T_{ii} = \sum_{i=1}^{k} w_{ui} T(\mathbf{x}_i), w_{ui} = \frac{p_i d_{ui}^{-2}}{\sum_{i=1}^{k} p_i d_{ui}^{-2}} \tag{7}$$

where $u$ and $i$ are the indices for users and reference locations, respectively. Each turbulence delay at the user position is assigned a weight $w_{ui}$, which is determined by the horizontal distance $d_{ui}$ from the user to the reference position, and the weight $P_i$ of GNSS and ECMWF, similar to the weighting mode of hierarchical delay in Equation (6) [18].

### 2.1.3. High-Resolution Numerical Atmospheric Model (ERA5) Correction Method

The ECMWF published ERA5, the fifth generation of meteorological reanalysis data, which provide a 31 km horizontal resolution from 1959 to the present. The meteorological reanalysis data have a relatively high temporal (1 h) [19] and spatial resolution. The barometric value, geopotential, relative humidity, and temperature variables of 37 pressure layers in the ERA5 meteorological reanalysis data grid node are obtained during the SAR imaging period and implement horizontal and vertical spline interpolation for pressure, temperature, and relative humidity. Meanwhile, to correspond with the SAR imaging time, linear interpolation is performed [2]. According to the incidence angle of the digital elevation model (DEM) and SAR images in the study area, the tropospheric delay of the interferogram in the radar line of sight is calculated by combining Equations (1)–(3), obtaining the tropospheric delay of each pixel in the interferogram in the radar line of sight.

## 2.2. Study Area and Data Processing

### 2.2.1. Overview of the Study Area and Data Sources

The study area is located in the middle reach of the Jinsha River (99°25′30″E–101°27′36″E, 27°08′42″N–28°02′38″N) of Lijiang City, Yunnan Province, in the middle of the Hengduan Mountains in southwest China [20]. The river runoff is abundant, river drop is high, and water flow is swift. Canyons are the main feature [21] of this low-latitude alpine region. The study area includes four climate types: plateau mountain, subtropical plateau monsoon, cold temperate, and subtropical monsoon. Deep inland, the terrain is complex, showing a dramatic rise from the central valley zone to the two sides of the river. The temperature decreases with the increase in altitude, alongside variable water conditions with a remarkable three-dimensional climate. Sentinel-1A ascending and descending data were processed over an area of 14,951.79 km$^2$, with a maximum elevation of 5353 m, minimum elevation of 1354 m, and an average elevation of 3094.80 m. As there are few GNSS monitoring stations in the study area, the key research area was delimited according to the overlapping areas of ascending and descending track data, as shown by the red range line in Figure 1. The blue and green range lines represent the InSAR processing areas of the ascending and descending tracks, respectively.

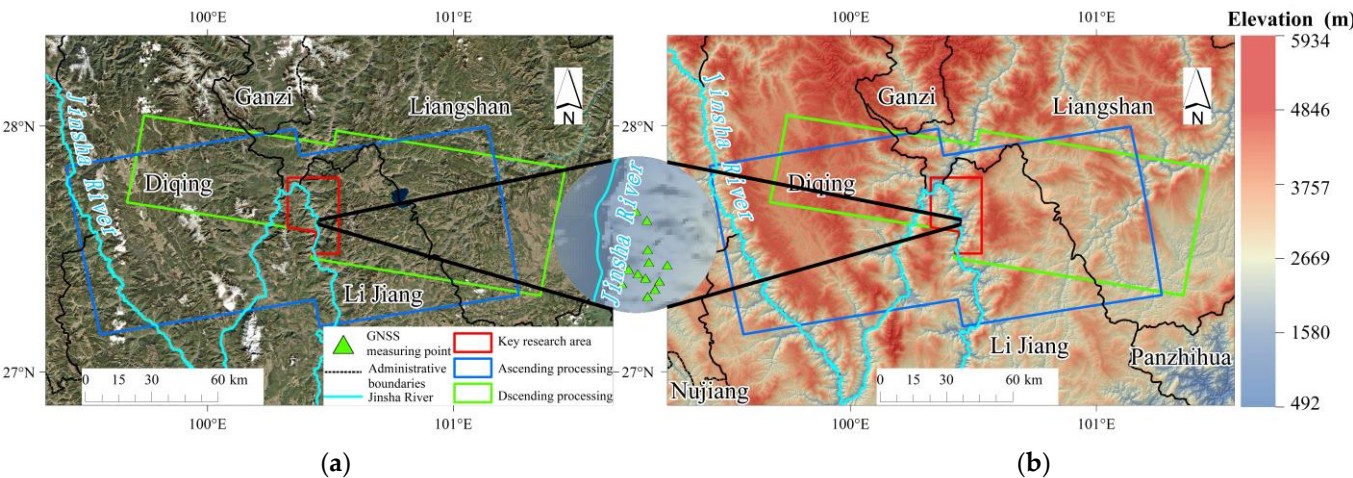

**Figure 1.** (**a**) Optical image and (**b**) topographic map of the study area in Lijiang City, Yunnan Province, China.

This study collected 24 C-band scenes and interferometric wide (IW) swath imaging mode Sentinel–1A ascending and descending track data of single complex (SLC) images (https://search.asf.alaska.edu/ accessed on 7 February 2023). The polarization mode was VV, and the data coverage period was from 17 June 2018 to 3 April 2019. DEM data were adapted from the NASA Shuttle Radar Topography Mission (SRTM) version 3.0 global 1-arc-second data with a spatial resolution of 30 m (https://search.earthdata.nasa.gov/search accessed on 7 February 2023). The GACOS ZTD products (http://www.gacos.net/ accessed on 7 February 2023) and ERA5 meteorological reanalysis data (https://cds.climate.copernicus.eu/cdsapp#!/search?type=dataset accessed on 7 February 2023) were retrieved from their respective websites.

### 2.2.2. Research Method

In this study, the empirical model (Linear method), GACOS method, and high-resolution atmospheric numerical model correction method (ERA5 method) were used for InSAR tropospheric delay correction. First, sequential InSAR processing was performed on the SLC data, to obtain the original interference phase sequence. As the geographical environment of the study area is mostly mountains and canyons with severe relief and a lack of surface objects with high coherence, SmallBaselineSubsetInSAR (SBAS-InSAR) technology [22] was adopted for sequential InSAR inversion. Second, the Linear, GACOS,

and ERA5 methods were used for tropospheric delay correction of the original interference phase sequence. Then, the tropospheric delay correction effect was evaluated using phase standard deviation ($STD_1$), semi-variant structure-function, and elevation correlation analysis of the statistically corrected interferogram sequences, without validation data. Finally, GNSS deformation monitoring data were used as true values to test the accuracy of InSAR deformation monitoring after tropospheric delay correction; the tropospheric correction method with the most suitable correction effect for the region was determined based on interferogram analyses. The technical process is illustrated in Figure 2.

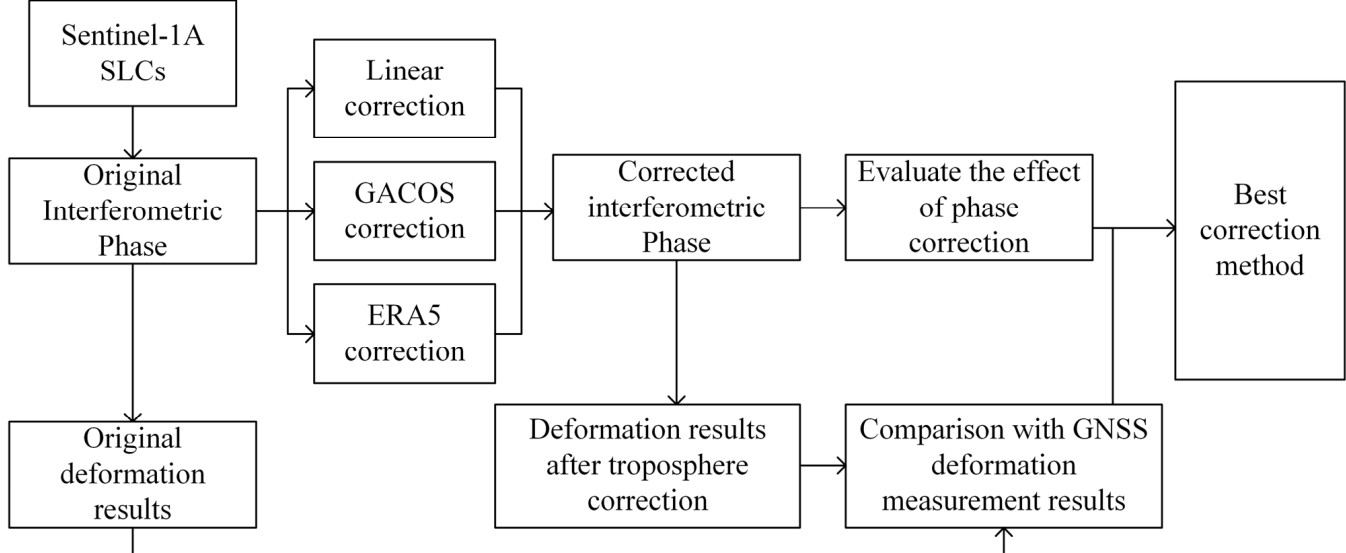

**Figure 2.** Technical flow chart of the research methodology.

The time baseline threshold was set to 135 d, and the maximum spatial baseline threshold was 180 m during the processing of the SBAS-InSAR time series. The processing range of the ascending track data covered two IW strips and four bursts, whereas the descending track covered two IW strips and three bursts. To suppress the noise and improve the signal-to-noise ratio of the image, multilooking processing of the range and azimuth directions was conducted with a multilooking number of 20 × 5. A total of 113 ascending-track and 112 descending track interference pairs were obtained. Fourteen interferograms with poor interference, caused by low coherence, were eliminated from the ascending and descending tracks through visual inspection. Finally, 99 ascending track interferograms and 98 descending track interferograms were used for timing inversion. The basic information of the images is shown in Table 1, and the space–time baseline is shown in Figure 3.

**Table 1.** Sentinel-1A data parameters.

| Tracks | Number of Images | Time Coverage | Beam Mode | Number of Initial Interferograms | Number of Final Interferograms |
|---|---|---|---|---|---|
| Ascending | 24 | 2018/06/17–2019/04/01 | IW | 113 | 99 |
| Descending | 24 | 2018/06/19–2019/04/03 | IW | 112 | 98 |

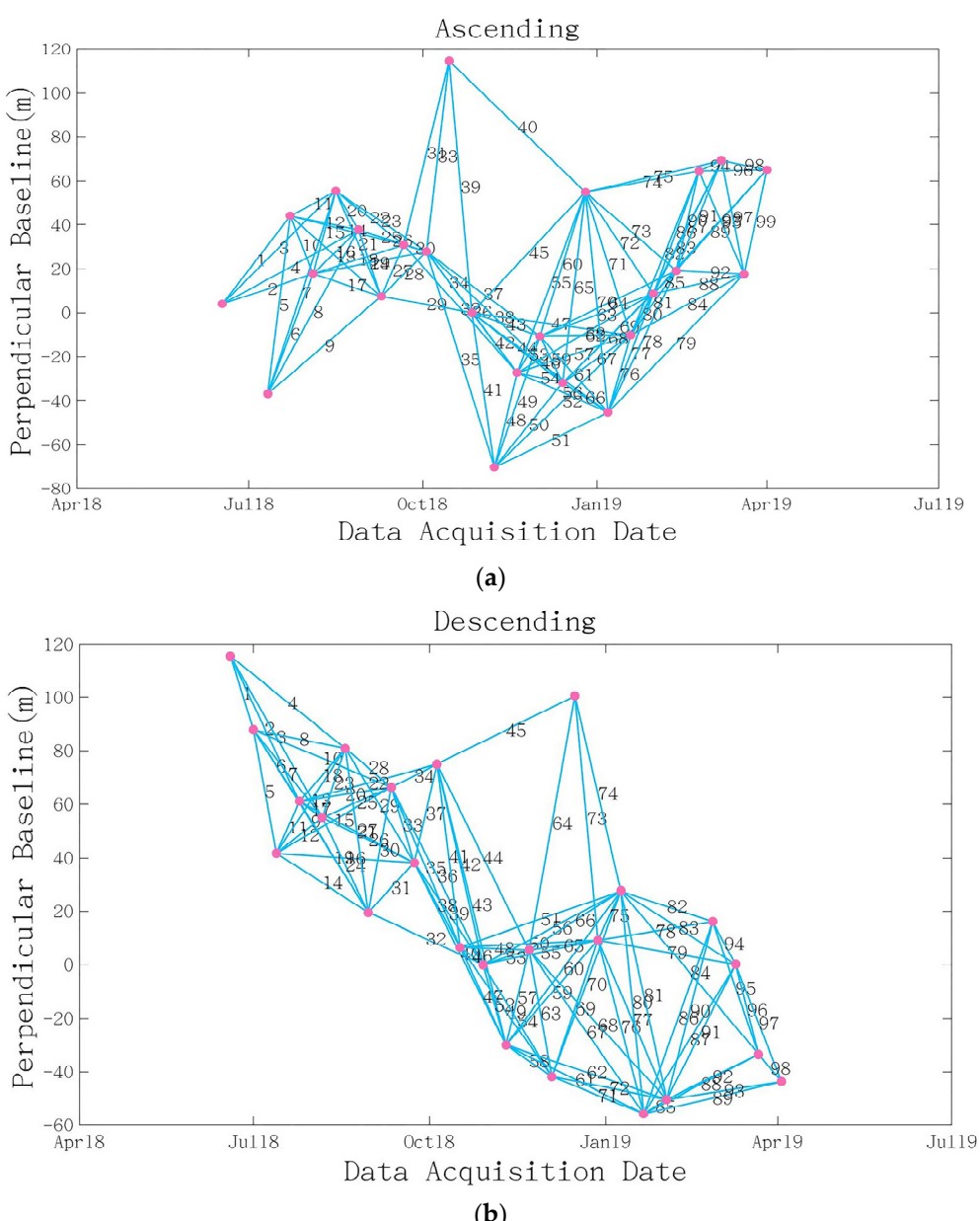

**Figure 3.** Space–time baseline map of (**a**) ascending tracks and (**b**) descending tracks.

## 3. Results

### 3.1. STD Evaluation

In this study, the Linear, GACOS, and ERA5 methods were each used to correct the tropospheric delay of 197 original interferograms in the ascending and descending tracks. The InSAR interference phase is composed of the reference ellipsoid, topographic, deformation, atmospheric, and noise phases [23]. Assuming that no deformation displacement occurs on the surface, after removing the reference ellipsoid, terrain, and noise phases in sequential InSAR technology, only the atmospheric phase remains; that is, tropospheric delay. If no surface deformation occurs, the phase standard deviation (STD) of the interference phase is used to evaluate the quality of the InSAR tropospheric correction [10,24]. The STD reflects the degree of dispersion of phase values in the interferogram. If it is larger in the non-deformation region, the tropospheric delay signal is more significant, and when it is reduced after correction for the tropospheric delay, the tropospheric delay is alleviated. In Equation (8), $\sigma$ is the standard deviation of interference phase of the interferogram (STD$_1$),

$N$ is the number of candidate phase points, and $x_i$ is the corresponding phase value of candidate phase points.

$$\sigma = \sqrt{\frac{1}{N} \sum_{i-1}^{N} (x_i - \mu)^2}, \mu = \frac{1}{N} \sum_{i=1}^{N} x_i \tag{8}$$

According to the group standard of the issued by the China Geological Disaster Prevention Engineering Association, an area with an absolute annual rate of SBAS-InSAR radial deformation (line-of-sight (LOS) direction) less than 5 mm/y without tropospheric delay correction is considered a non-subsidence area [25] or non-deformation area. The interference phases of the non-deformation regions were counted, and the mean value of the interference phase standard deviation (Aver) of all interferograms corresponding to the ascending and descending tracks was calculated, as shown in Table 2. The Aver of the 99 original interferograms of ascending tracks without tropospheric correction was 2.86 rad, and the mean standard deviation of the interference phase (Aver) of the Linear, GACOS, and ERA5 methods decreased to 2.26, 2.43, and 2.40 rad, respectively. The Aver of the 98 original interferograms of the descending tracks was 1.85 rad, and the Aver of the Linear, GACOS, and ERA5 methods decreased to 1.55, 1.67, and 1.65 rad, respectively. Overall, the three correction methods effectively reduced the interference phase standard deviation ($STD_1$) of the original interferogram in the non-deformation region and alleviated the tropospheric delay in the original interferogram. After tropospheric correction of all ascending and descending track interferograms, the Aver corresponding to the Linear method decreased the most. Among the correction methods, the GACOS method decreased the least.

**Table 2.** Variation of the average value of interference phase of all interferograms.

| Track | Evaluation Indicators | Original | Linear | GACOS | ERA5 |
|---|---|---|---|---|---|
| Ascending | Average of standard deviation of interference phase of all IFGs (Aver)/rad * | 2.86 | 2.26 | 2.43 | 2.40 |
| | Rate of change | - | −20.98% | −15.03% | −16.08% |
| Descending | Average of standard deviation of interference phase of all IFGs (Aver)/rad | 1.85 | 1.55 | 1.67 | 1.65 |
| | Rate of change | - | −16.22% | −9.73% | −10.81% |

\* IFGs are interferograms, and Aver is the average of the standard deviation of interference phase ($STD_1$) of all interferograms.

Not all results of the tropospheric delay correction for the original interferogram were valid. After correction, the $STD_1$ of a single interferogram increased, which was an unsatisfactory correction result, called overcorrection. Overcorrection occurred in both ascending and descending track interferogram corrections, as shown in Figures 4 and 5. Table 3 shows the number of tropospheric positive corrections or overcorrections in the interferogram. The $STD_1$ of 85 of the 99 interferograms in the ascending tracks decreased after the Linear correction method, with an average decline rate of −20.79%. The $STD_1$ of the 14 other interferograms increased at an average rate of 2.90%. After correction using the GACOS method, the $STD_1$ of 70 interferograms decreased, at an average of −19.62%, while the other 29 interferograms increased with an average increase of 8.97%. After correction using the ERA5 method, the $STD_1$ of 75 interferograms decreased by −18.69%, while 24 interferograms increased by 7.14%.

Among the 98 interferograms of descending tracks, the $STD_1$ of 67 interferograms decreased after Linear correction, while the $STD_1$ of the 31 other interferograms increased, with an average rate of 10.11%. After correction by the GACOS method, the $STD_1$ of 56 interferograms decreased, with an average rate of –19.50%, whereas the other 42 interferograms increased by

18.21%. After correction using the ERA5 method, the $STD_1$ of 52 interferograms decreased, with an average decrease rate of −22.29%, and the $STD_1$ of 46 interferograms increased, with an average increase rate of 15.11%. Figure 5 shows that, after correction, the reduction in the $STD_1$ of the original interferogram of the ascending tracks was within the interval (−60%, 0%). The Linear method could reasonably alleviate the tropospheric delay of the interferogram, with the best performance being 85.86%, followed by the ERA5 method, which could alleviate 75.76% of the tropospheric delay of the interferogram. The reduction in the $STD_1$ of the descending track was in the range (−70%, 0%), while the Linear method could relieve 68.37% of the tropospheric delay of the interferogram, followed by the GACOS method, which could relieve 57.14% of the tropospheric delay of the interferogram. Overall, the correction effect for the ascending track interferogram was better than that for the descending track interferogram.

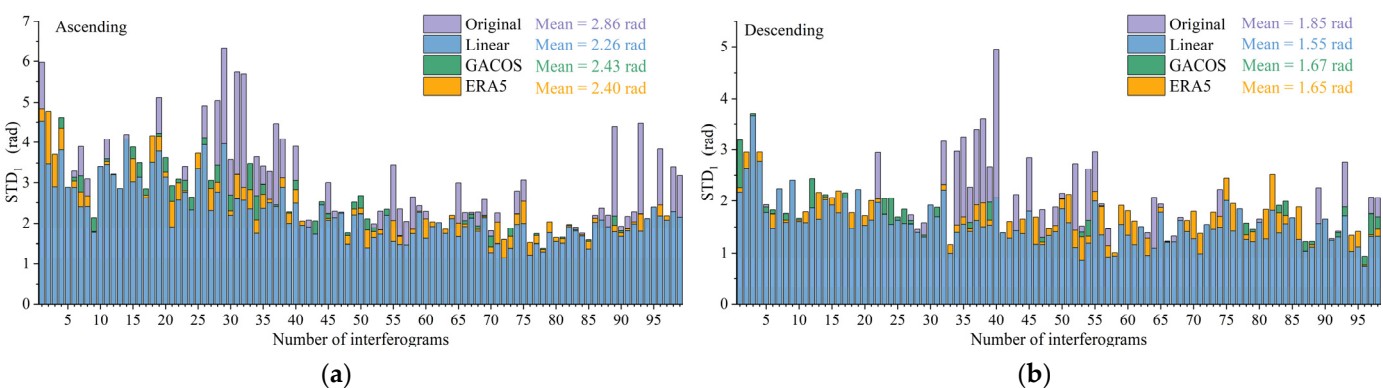

**Figure 4.** Statistical diagram of the standard deviation of interference phase ($STD_1$) before and after tropospheric correction of (**a**) 99 interferograms in ascending tracks; (**b**) 98 interferograms in descending tracks.

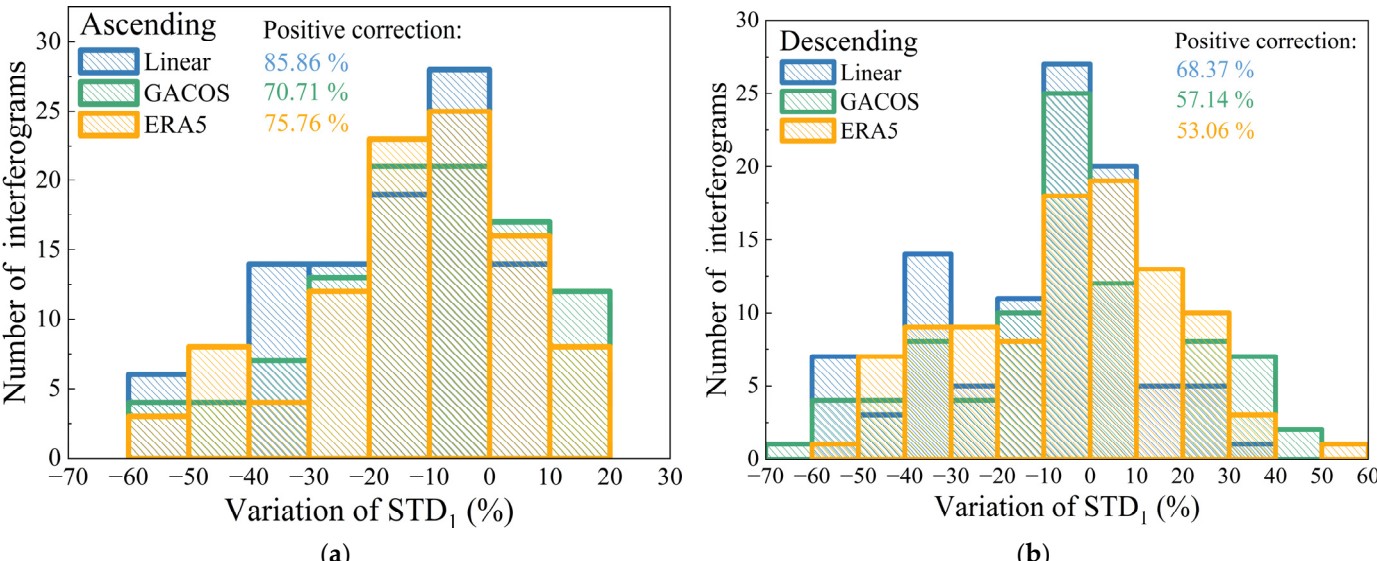

**Figure 5.** Histogram of variation of interference phase standard deviation ($STD_1$) of (**a**) ascending track interference phase; and (**b**) variation of the standard deviation ($STD_1$) of descending track interference phase.

**Table 3.** $STD_1$ change of average interference phase of all interferograms.

| Track | Evaluation Indicators | Linear | GACOS | ERA5 |
|---|---|---|---|---|
| Ascending | Number of IFGs increased/decreased by $STD_1$ | +14/−85 [1] | +29/−70 | +24/−75 |
| | Rate of change of the average value of $STD_1$ | +2.9%/−20.79% [2] | +8.97%/−19.62% | +7.14%/−18.69% |
| | Increase/Decrease of the average value of $STD_1$ | +0.07/−0.72 rad [3] | +0.21/−0.70 rad | +0.17/−0.67 rad |
| Descending | Number of IFGs increased/de-creased by $STD_1$ | +31/−67 | +42/−56 | +46/−52 |
| | Rate of change of the average value of $STD_1$ | +10.11%/−21.09% | +18.21%/−19.50% | +15.11%/−22.29% |
| | Increase/Decrease of the average value of $STD_1$ | +0.15/−0.51 rad | +0.27/−0.50 rad | +0.22/−0.56 rad |

[1] "+14" represents 14 interferograms of rising $STD_1$, "−85" represents 85 interferograms of falling $STD_1$. [2] "+2.9%" represents the average growth rate of interference phase $STD_1$ corresponding to 14 interferograms with $STD_1$ rising, "−20.79%" represents the average decline rate of interference phase $STD_1$ corresponding to 85 interferograms with $STD_1$ falling. [3] "+0.07 rad" represents the average increase of interference phase $STD_1$ corresponding to 14 interferograms with $STD_1$ rising, "−0.72 rad" represents the average decrease of interference phase $STD_1$ corresponding to 85 interferograms with $STD_1$ falling, and so on.

After correction, the number of interferograms with correction in ascending and descending tracks was less than 50%, and the average increase in $STD_1$ was <0.3 rad, which could be ignored because the value was small. After correction, the Aver decreased. All three methods positively affected the correction. As can be seen from Figure 4, compared with the $STD_1$ of the original interferogram, most of the interferograms of the ascending and descending tracks decreased after the correction of the three methods, and the degree of reduction after the correction of the linear method was more obvious than that of the GACOS and ERA5 methods. The change in the $STD_1$ of the interferograms of the three correction methods was compared in pairs, as shown in Figure 5. The upper right of each figure shows the overcorrected interferogram, and the linear method reduced the $STD_1$ of most interferograms. This method was clearly superior to the GACOS and ERA5 methods in the interferogram of ascending and descending tracks, and the degree of correction of the GACOS and ERA5 methods was roughly similar. Based on Aver, the increase and decrease of the $STD_1$ of a single interferogram, and the mean change of standard deviation (as shown in Tables 2 and 3, Figures 4 and 6), the linear correction method showed the best improvement of the tropospheric delay of the ascending and descending tracks, and the Aver of all interferograms decreased. The correction effects of the GACOS and ERA5 methods were similar, with no significant differences. In addition, the imaging time of the descending tracks SAR image data was 23:13 coordinated universal time (UTC), which is equivalent to 7:13 a.m. china standard time (CST), and the imaging time of the ascending tracks SAR image data was 11:24 UTC, which is equivalent to 19:24 CST at night. Usually, the temperature at 19:00–20:00 CST at night is higher than that at 7:00–8:00 CST in the morning, and the molecular activity frequency in the atmosphere at the time of descending track SAR image imaging was lower than that at night, the atmosphere was more stable, and so the tropospheric delay of the descending tracks interferogram was shorter. It was difficult for the three correction methods to correct the smaller scale tropospheric delay, so the number of descending track overcorrected interferograms was higher than ascending track the interferograms.

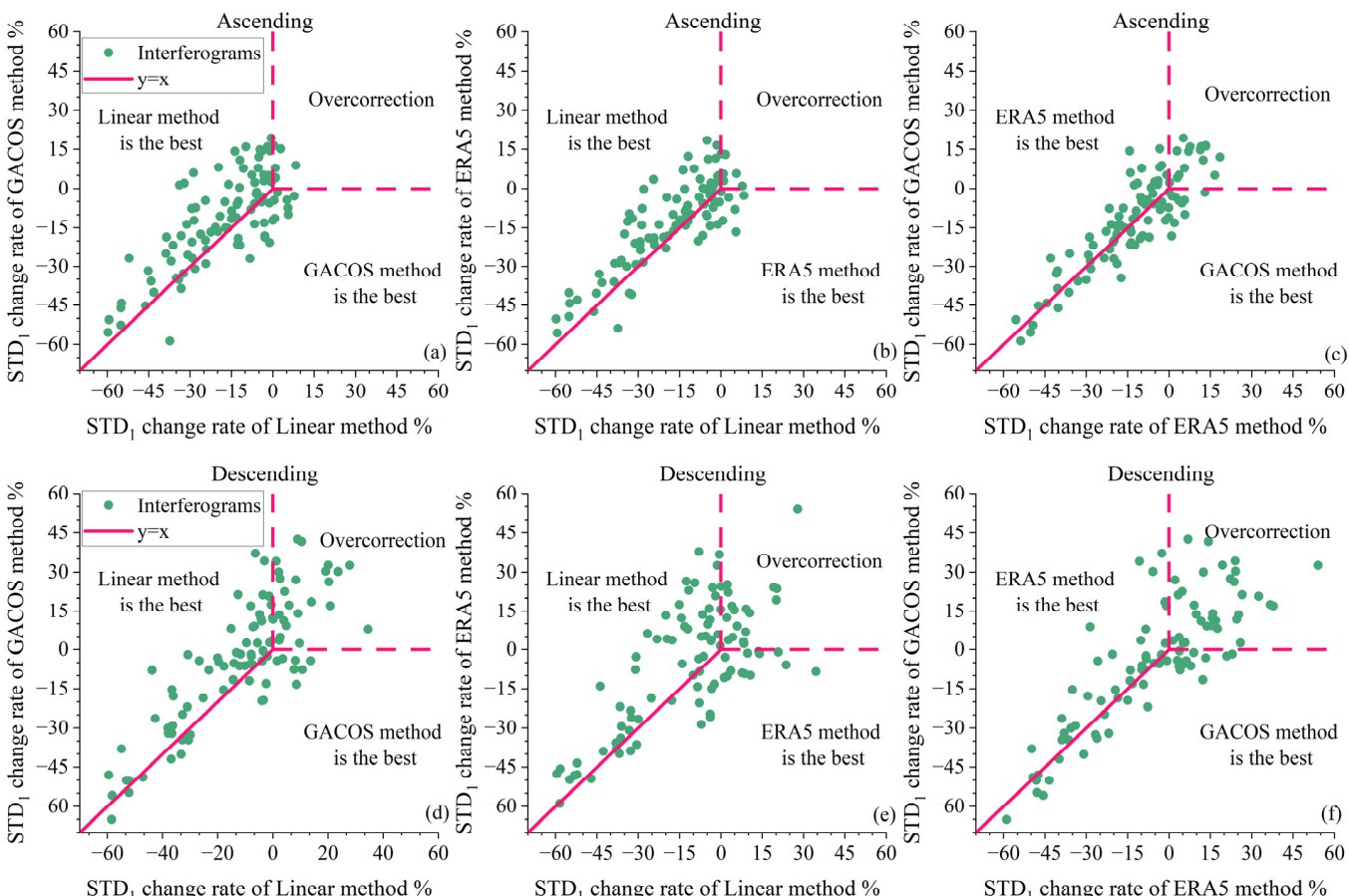

**Figure 6.** Comparison of tropospheric delay correction methods; (**a–f**) show the change comparison of the standard deviation of the interference phase (STD$_1$) of the three correction methods for the ascending and descending track interferograms. The green dot represents the interferogram, the negative change rate of the STD$_1$ indicates that the tropospheric delay was alleviated, the positive value represents the overcorrection and the pink realization, and the dashed line represents the dividing line between the optimal correction method and the overcorrected interferogram.

### 3.2. Semi-Variance Function Evaluation

The tropospheric signals exhibited spatial variability. The turbulent process of atmospheric water vapor led to an inconsistent spatial distribution of the atmospheric refractive index in the two observation periods, and the delay of various positions at the same height differed in space [26]. Tropospheric signals with long wavelengths manifested as stratified or turbulent components related to topography or existed simultaneously [18]. STD$_1$ does not analyze the correlation between tropospheric delay, distance, and elevation, and cannot reflect the changes in the spatial structure dimension of tropospheric delay after correction. Other indices should be introduced to evaluate the spatial variation characteristics of tropospheric delay. The semi-variance function contains useful information about a specific spatial scale. At a specific spatial scale, tropospheric delay produces a positive correction or overcorrection effect, and the structural function can separate the correction effects at different spatial scales [5]. In Equation (9), $\gamma(h)$ is the semi-variance within the interval of hysteretic distance $h$, $N(h)$ is the total number of candidate interference phase point pairs within hysteretic distance $h$, $z_i$ is the interference phase value at the position of point $i$, and $z_{i+h}$ is the interference phase value at point $i + h$.

$$\gamma(h) = [1/2N(h)] \sum [z_i - z_{i+h}]^2 \tag{9}$$

The semi-variance value of the function increases with increasing sample spacing and tends to be stable at a certain distance. This distance is called the out-of-correlation distance (range), and the semi-variance value corresponding to it is the threshold value (sill). The long-wave atmospheric signal was range-dependent and the out-of-correlation distance reflected the autocorrelation distance scale of the spatial data. Outside the out-of-correlation distance, the sample data were not correlated; whereas, within the out-of-correlation distance, the sample data had a spatial correlation. After tropospheric correction of the interferogram, the sill decreased, indicating that the tropospheric delay noise level in the interferogram had decreased and the long-wave signal had weakened [10]. To improve the computational efficiency, 5% of the sample points were randomly extracted from the non-deformation region of the interferogram of the ascending and descending tracks before and after correction, and these sample points were evenly distributed in the non-deformation region. Since the sample data were obtained by random sampling, the lag distance was generally set to about half of the maximum distance of the sample points. We set the ascending track lag distance to 93.372 km with a step size of 6.224 km, and a descending track lag distance of 86.382 km with a step size of 5.758 km. The range and sill of the pre-and post-correction interferograms were calculated using an exponential model. As shown in Table 4 and Figure 7, after tropospheric correction, the mean sill and semi-variance of the interference phase of all interferograms decreased, and the three correction methods all reduced the tropospheric noise level in the interferogram. Among them, the mean sill of the Linear correction method decreased the most, similar to the reduction in the Aver of the Linear correction method (Table 2). However, for the mean loss of correlation distance (range) of all interferograms, that of the Linear correction method was larger than that of the original interferogram, indicating that the interference phase of the interferogram corrected using the Linear method was highly correlated at a certain spatial scale, because the turbulent component of tropospheric delay cannot be estimated using the Linear correction method. A turbulent mixing delay still existed in the interferogram. After correction using the ERA5 method, the mean loss of range of the ascending tracks increased, while the for the descending tracks it decreased. This indicated that the ERA5 method can capture the turbulence delay in space and correct long-wave tropospheric signals, but its low spatial resolution may not produce a positive correction effect. Inversely, GACOS products benefit from the characteristics of a high spatial resolution. After correction, the mean loss of range value of the ascending and descending track interferograms decreased, as well as the spatial scale of the interference phase correlation. From the perspective of the spatial distribution characteristics of tropospheric delay, the ability of the GACOS method to correct the long-wave signal in space was better than that of the Linear and ERA5 methods.

**Table 4.** Statistical table of sill and range changes before and after tropospheric correction.

| Tracks | Type | Original | Linear | GACOS | ERA5 |
|--------|------|----------|--------|-------|------|
| Ascending | Average value of sill (rad$^2$) | 0.087 | 0.051 | 0.058 | 0.058 |
| | Average value of range (km) | 55.361 | 77. 785 | 46.867 | 67.567 |
| Descending | Average value of sill (rad$^2$) | 0.056 | 0.035 | 0.046 | 0.052 |
| | Average value of range (km) | 92.651 | 93.569 | 63.649 | 88.179 |

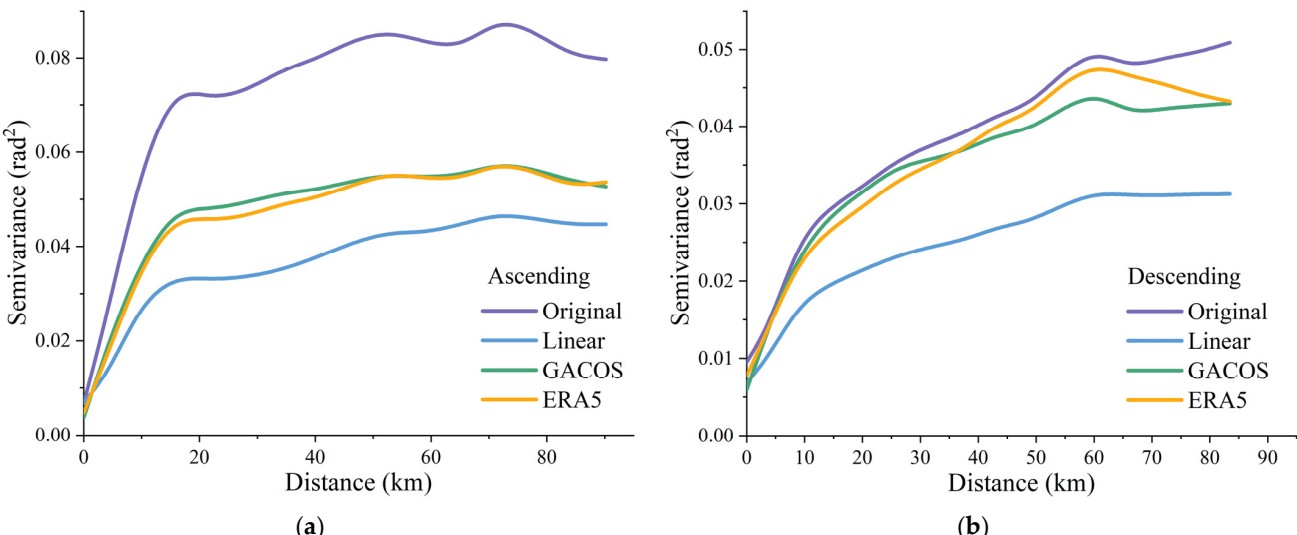

**Figure 7.** Semi-variogram of the mean value of semi-variance of (**a**) ascending and (**b**) descending track interferograms before and after troposphere correction.

### 3.3. Elevation Correlation Evaluation

Tropospheric delay is closely related to elevation change, and varied atmospheric refractive indices at different altitudes lead to a stratification effect. To analyze the relationship between the interference phase and elevation in the interferogram, their relationship was established. The Pearson correlation coefficient measures the correlation between two variables. The closer the correlation coefficient is to 1 or –1, the stronger the correlation between two variables; the closer the correlation coefficient is to 0, the weaker the correlation between variables. Owing to the numerous interferograms, we selected one with two days' separation between the ascending and descending tracks and extracted the common part of the ascending and descending track interferograms, to analyze and compare the relationship between the interference phase and elevation.

In Figure 8, the ascending and descending tracks had two days before and after their acquisition date, both from September to October. During this period, the time interval was short and they were in the same season. The deformation and seasonal differences between the two ascending and descending track interferograms can be ignored. From the original interference phase value and Pearson correlation coefficient, the correlation between the interference phase and elevation is different, even in the same region, owing to changes in atmospheric conditions at a similar time. The original interference phase value included the deformation and the tropospheric delay phases. The Pearson correlation coefficients $r$ between the original interference phase and the altitudes of the ascending and descending tracks were 0.87189 and 0.90126, respectively. The interference phase had a strong correlation with the altitude. The linear correction method was far more sensitive to the vertical stratification delay than the GACOS and ERA5 methods, which significantly weakened the vertical stratification delay. The Pearson correlation coefficients of the ascending and descending tracks were reduced to 0.093356 and 0.34484, respectively, and the interference phase was either very weakly correlated with the elevation or had no correlation. After the three methods were used to remove the tropospheric delay phase, the correlation between the interference phase and elevation decreased significantly, indicating that the three methods weakened the vertical stratification component of the tropospheric delay and that it occupied a dominant position in this region. However, the ability of GACOS and ERA5 to capture the vertical stratification component was poor. In similar periods, the weakened performance for the vertical stratification delay was unstable, and the mitigation effect was different. The air in the troposphere mostly moves in a vertical direction and can be divided into lower (below 1500 m height above the ground), middle (about 1500–6000 m), and upper (about 6000–11,000 m) layers, according to air currents

and weather phenomena. The cold air in the upper layer always has a sinking trend and the warm air in the lower layer always has a rising trend [27]. The correlation coefficient r decreases differently between GACOS and ERA5 methods in the ascending and descending interferograms. We speculate that the descending tracks SAR images were imaged in the morning, CST time, when the lower troposphere temperature is lower and the degree of vertical motion is slower, and the GACOS and ERA5 methods were less capable than the Linear method of correcting the vertical stratification delay at smaller scale.

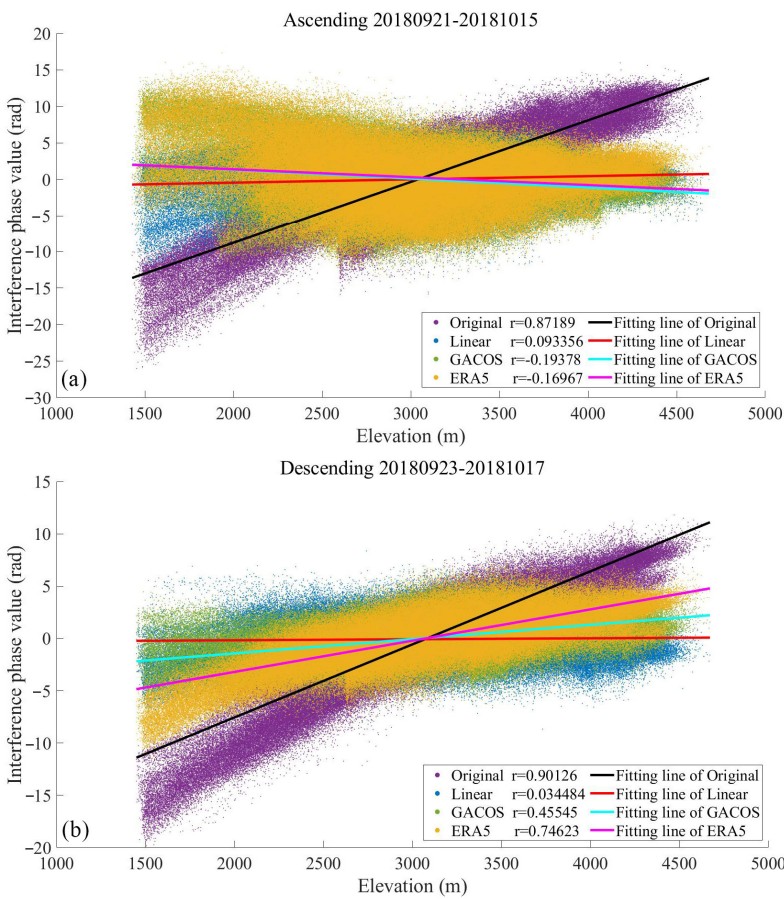

**Figure 8.** Relationship between interference phase and elevation: Correlation between interference phase and elevation of (**a**) ascending tracks from 21 September 2018–15 October 2018 and (**b**) descending tracks from 23 September 2018–17 October 2018.

### 3.4. GNSS Station Deformation Monitoring and Evaluation

Figure 9 shows the InSAR deformation rate results before and after the tropospheric delay correction. Positive values indicate that the target moved closer to the satellite in the LOS direction, whereas negative values indicate that the target moved away. Compared with the original deformation rate results, that of the InSAR inversion were changed by the three correction methods, with significantly different ranges. The difference in the maximum deformation rate was 12.2 mm/y. According to the ascending and descending track data, owing to the distribution of rivers on both sides of the Jinsha River basin and the high topographic drop, the changes in water vapor in the air were more drastic, and the changes in the InSAR deformation rate were more obvious. In the original deformation rate results, some regions with high deformation rates exhibited disordered stains and excessive noise. After tropospheric correction, the region with a high deformation rate for the ascending track data showed an expanding trend, while that of the descending track data showed a decreasing trend. The deformation stains became more obvious, distribution became more uniform, and the transition became smoother. The noise in the deformation rate results was largely alleviated.

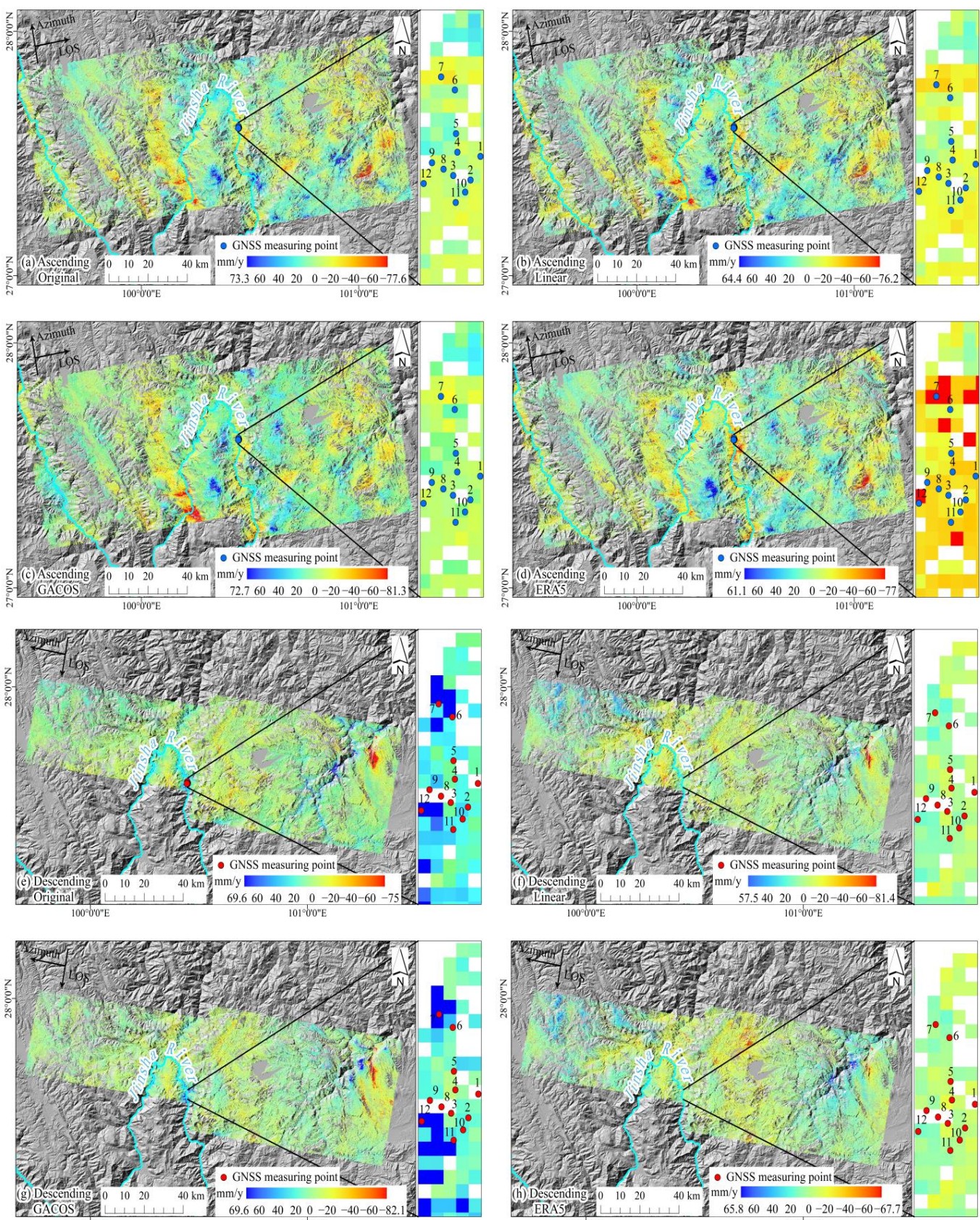

**Figure 9.** InSAR deformation rate results before and after tropospheric delay correction in (**a**–**d**) ascending tracks and (**e**–**h**) descending tracks.

InSAR deformation monitoring of a time series is a process in which the surface displacement is obtained using the accumulation of the interferogram phase within a period of time. To more effectively evaluate the variation in InSAR deformation monitoring accuracy before and after tropospheric correction, the three-dimensional deformation monitoring results of 12 GNSS stations in the east–west, north–south, and vertical directions (NEU) were projected into the LOS direction according to Equation (10) [28]. The one-dimensional displacement of the GNSS three-dimensional deformation in the LOS direction was obtained, consistent with the reference of the InSAR. These 12 GNSS stations were monitored from 29 June 2018 to 6 April 2019, and the average root mean square error (RMSE) of the points was 1.9 mm, and the average geodesic height RMSE was 6.5 mm, which had a high measurement accuracy and ensured the reliability of the validation data in this study. Since the monitoring method was a static measurement with a GNSS receiver, GNSS monitored a total of five periods with six time nodes, and InSAR monitored a total of 23 periods with 24 time nodes, and the node dates were not fully consistent. We compared InSAR with the cumulative deformation values of the six nodes with the closest dates in the GNSS deformation time series, and calculated the average RMSE of InSAR at each GNSS site for the six node dates, before and after tropospheric correction. The more the InSAR time series node dates matched with GNSS time series node dates, the more accurate the deformation monitoring values and deformation trends. Although there was a slight difference between the deformation timing node dates of InSAR and GNSS, this did not affect the validity of our study. After tropospheric delay correction, as long as the average RMSE is reduced, this indicates that the tropospheric delay error is reduced.

$$d_{\text{GNSS}} = \sin\theta \cdot \sin(a) \cdot dx - \sin\theta \cdot \cos(a) \cdot dy + \cos\theta \cdot dz \quad (10)$$

where $d_{\text{GNSS}}$ is the shape variable projected by GNSS in the LOS direction; $a$ represents the azimuth of the SAR satellite heading vector (positive clockwise from north); $\theta$ is the incidence angle of the radar wave; and $d_x$, $d_y$, and $d_z$ represent the north, east, and vertical deformation values of the GNSS stations, respectively.

The deformation value of the GNSS was projected back to the LOS, and the relationship between InSAR and GNSS was derived. As shown in Figure 10, when the deformation gradient was small, the GNSS deformation value and trend were consistent with the InSAR deformation monitoring results. The difference between the descending track InSAR deformation value and GNSS monitoring deformation value was large. This may have been because of the high deformation value projected by the GNSS in the LOS direction of the descending tracks, the absolute value of deformation being generally greater than 100 mm, and the high deformation gradient near the GNSS point. This is far beyond the SBAS-InSAR deformation monitoring range of 1 cm/y–1 dm/y [25]. However, as the deformation trend was highly consistent, it did not affect the accuracy of the tropospheric delay correction. The three tropospheric correction methods generally improved the accuracy of InSAR deformation monitoring, among which the monitoring accuracy of sites 2, 7, 10, 11, and 12 in the ascending tracks and sites 4, 5, 6, 8, 11, and 12 in the descending tracks were greatly improved. However, in the vicinity of some GNSS sites, tropospheric delay correction seemed to produce an overcorrection effect, and the deformation monitoring accuracy of InSAR decreased, which also demonstrated that tropospheric conditions in different scale-spaces can have significant differences within the same time period.

According to the analysis in Table 5, among the ascending track data, the Linear method had the best tropospheric correction effect, and the mean value of RMSE was the lowest, at 55.2 mm, followed by the GACOS method, at 63.6 mm. The ERA5 method could improve the deformation monitoring accuracy of some stations, owing to its greater overcorrection effect. The mean value of RMSE increased from 65.5 mm to 68.3 mm, which also proved that the correction effect of the ERA5 method in different regions fluctuated greatly and had poor stability. For the descending track data, the average RMSE of the Linear, GACOS, and ERA5 methods decreased from 97.6 mm to 90.0 mm, 85.2 mm, and 90.5 mm, respectively. The average RMSE of the GACOS method was the lowest, followed

by that of the Linear and ERA5 methods. The mean RMSE values of the three methods decreased. As tropospheric conditions differ greatly in space, GNSS monitoring stations in the study area have a small distribution range and the maximum distance between GNSS monitoring stations is only 990.7 m; hence, the verification of measured data only represented the verification results of tropospheric correction at a small spatial scale.

**Table 5.** Root mean square error (RMSE) before and after tropospheric correction.

| | Ascending | | | | | Descending | | | |
|---|---|---|---|---|---|---|---|---|---|
| GNSS Site | RMSE of the Original Method | RMSE of Linear Method | RMSE of GACOS Method | RMSE of ERA5 Method | GNSS Site | RMSE of the Original Method | RMSE of Linear Method | RMSE of GACOS Method | RMSE of ERA5 Method |
| 1 | 64.9 * | 50.8 | 60.9 | 65.5 | 1 | 90.4 | 90.5 | 88.2 | 92.5 |
| 2 | 45.7 | 19.5 | 24.4 | 25.6 | 2 | 108.4 | 115.4 | 118.6 | 112.0 |
| 3 | 87.8 | 85.3 | 94.8 | 99.3 | 3 | 211.3 | 216.6 | 217.6 | 215.3 |
| 4 | 75.1 | 72.0 | 80.2 | 86.9 | 4 | 103.3 | 95.2 | 90.6 | 94.3 |
| 5 | 80.9 | 78.3 | 87.2 | 94.4 | 5 | 92.4 | 84.1 | 73.7 | 84.9 |
| 6 | 47.0 | 43.3 | 48.1 | 57.7 | 6 | 77.8 | 67.1 | 65.9 | 61.3 |
| 7 | 51.6 | 31.7 | 40.7 | 44.4 | 7 | 20.8 | 21.4 | 22.3 | 19.2 |
| 8 | 79.0 | 81.4 | 89.4 | 97.0 | 8 | 242.4 | 238.9 | 232.6 | 241.6 |
| 9 | 93.6 | 95.4 | 107.4 | 112.1 | 9 | 120.3 | 134.6 | 134.6 | 134.0 |
| 10 | 57.6 | 43.1 | 53.5 | 56.7 | 10 | 75.5 | 80.7 | 83.3 | 78.2 |
| 11 | 48.3 | 28.1 | 33.0 | 33.5 | 11 | 51.2 | 45.7 | 40.6 | 49.7 |
| 12 | 54.6 | 33.6 | 43.3 | 46.3 | 12 | 18.3 | 9.2 | 7.6 | 11.2 |
| Mean | 65.5 | 55.2 | 63.6 | 68.3 | Mean | 97.6 | 90.0 | 85.2 | 90.5 |

* Values in the table are in millimeters.

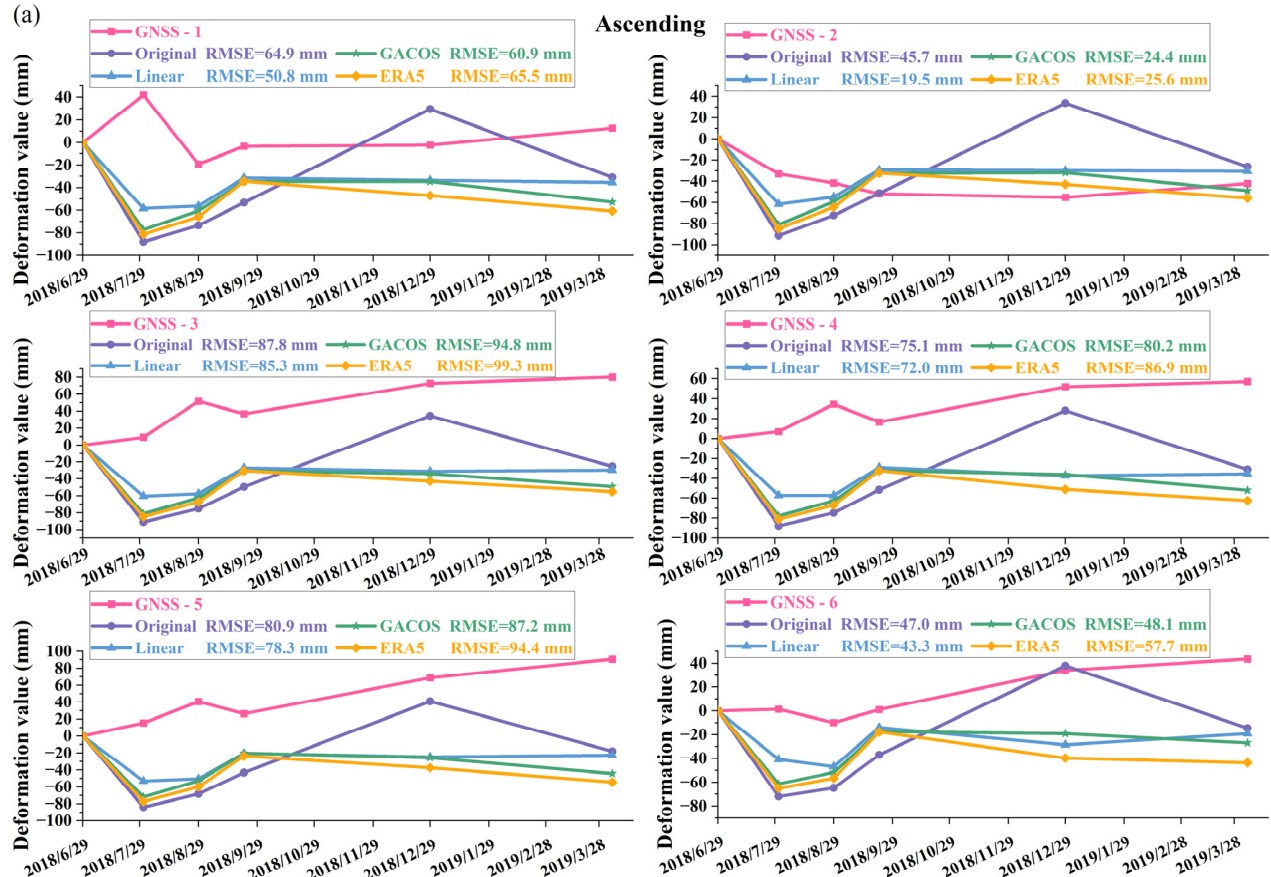

**Figure 10.** *Cont.*

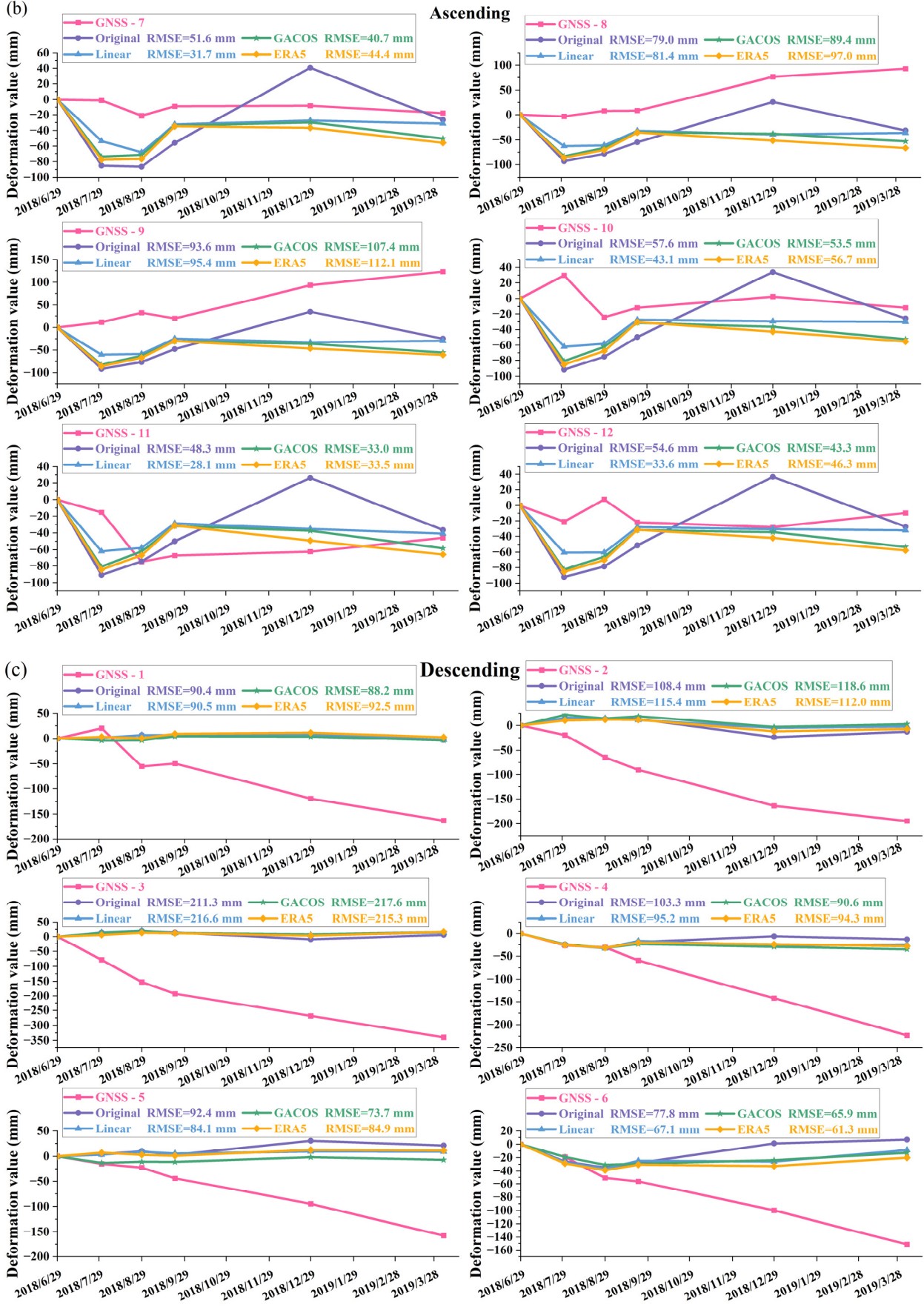

**Figure 10.** *Cont.*

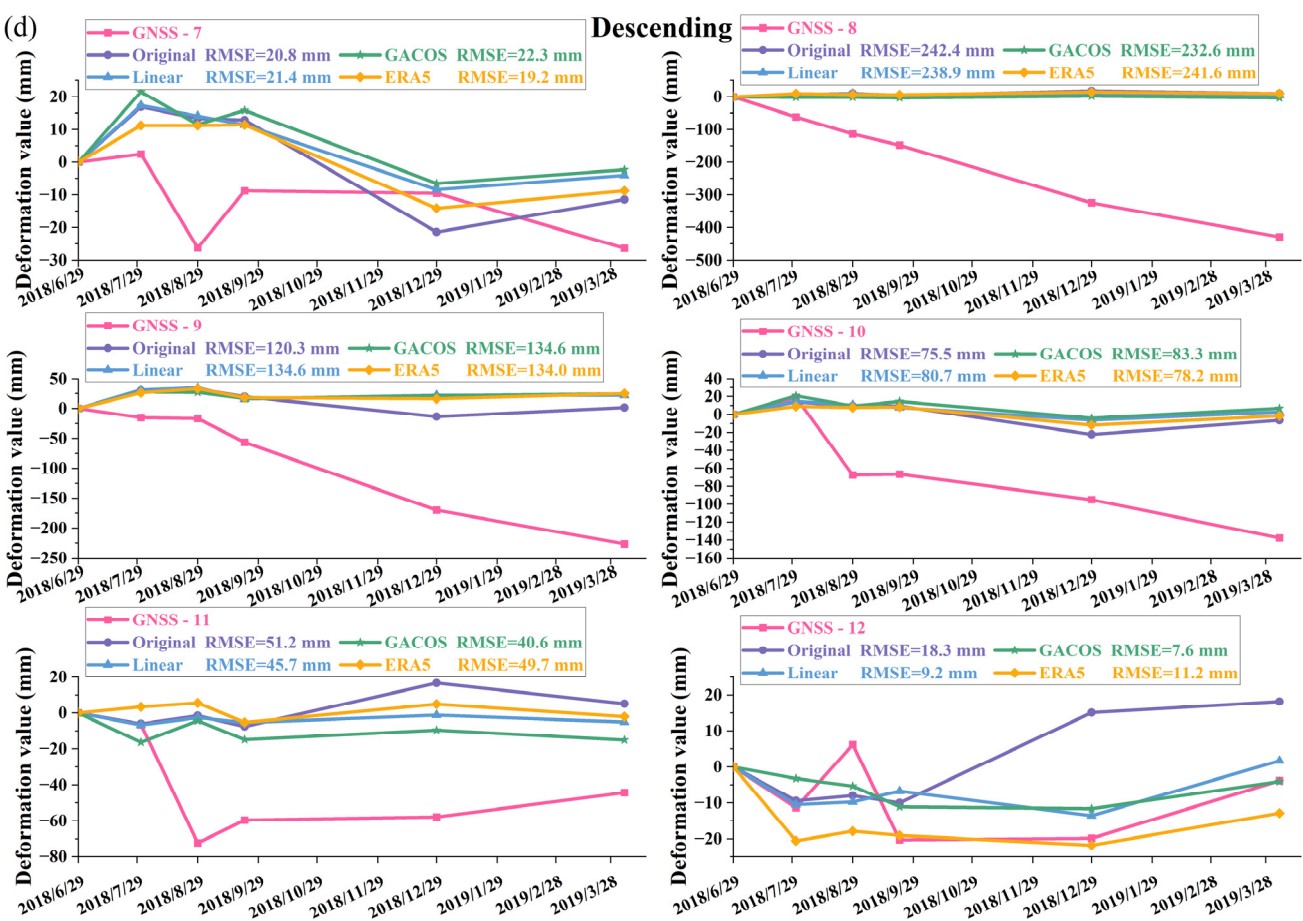

**Figure 10.** Comparison of deformation values between InSAR and GNSS before and after tropospheric correction of (**a**,**b**) ascending track data and (**c**,**d**) descending track data.

## 4. Discussion

The study area includes a variety of climate types, large topographic fluctuations, atmospheric temperature, water vapor, pressure, and other conditions that continually change, resulting in significant differences in the tropospheric delay in space and time scales. The tropospheric delay values were both positive and negative. To explore the seasonal variation of the tropospheric delay, only its value was considered, without the positive and negative relationships; and the average of the absolute value of the tropospheric delay estimated by the three interferogram methods was calculated. We divided March–May of a year into spring, June–August into summer, September–November into autumn, and December–February of the following year into winter. The average absolute values of the tropospheric delay estimated by the three methods are shown in Figure 11. The background color green represents the acquisition time of the original interferogram data in summer, orange represents autumn, and purple represents winter. The pink represents spring, and the colors alternate with each other, indicating that the original interferogram data were obtained in two different seasons. In Figure 11, the tropospheric delay estimated by the three methods has a high consistency, among which the mean change trends of the tropospheric delay estimated by the GACOS and ERA5 methods were more similar, because their data sources both used the meteorological data model for initial estimation.

When the original data acquisition period of any scene of the interferogram was in winter, the average value of the absolute tropospheric delay of the ascending and descending track interferograms was smaller than that of the other seasons, indicating that the atmospheric conditions in this region were relatively stable in winter. However, when any field image in the interferogram was in spring, summer, or autumn, the average

absolute value of the tropospheric delay was high. Interferograms with an increase in the $STD_1$ had their original image acquisition time in these three seasons. Figure 12 shows the temperature and total precipitation values of the ERA5 meteorological reanalysis data at 11:00 UTC, the closest to the UTC time of 11:24 of the ascending track SAR image. In winter, the temperature and total precipitation from 14 December 2018 to 31 January 2019, were significantly lower than in other seasons, and the total precipitation was almost 0 mm during most of the winter. In spring, summer, and autumn, the temperature, surface water vapor evaporation, and precipitation were all higher than that in winter. The tropospheric activity was also more higher. The tropospheric conditions at the time of SAR image acquisition were quite different, leading to the tropospheric delay value in the interferogram being too high and prone to overcorrection. For InSAR deformation monitoring of long time series, if the number of images allows, using SAR images acquired in winter for time series inversion is recommended, because the atmospheric conditions are stable during this time period and the image is less affected by tropospheric activities.

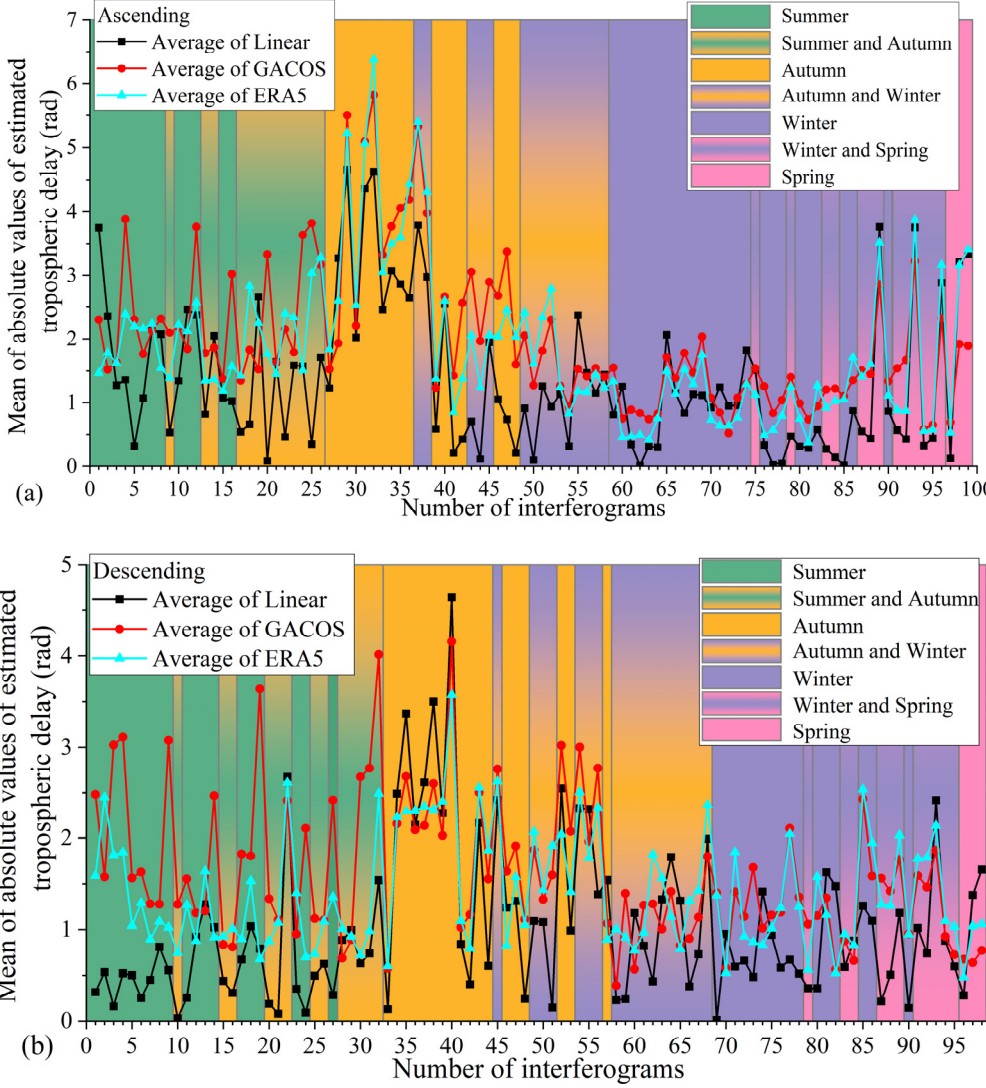

**Figure 11.** Mean of absolute value of the tropospheric delay estimated using three methods for (**a**) ascending and (**b**) descending track data.

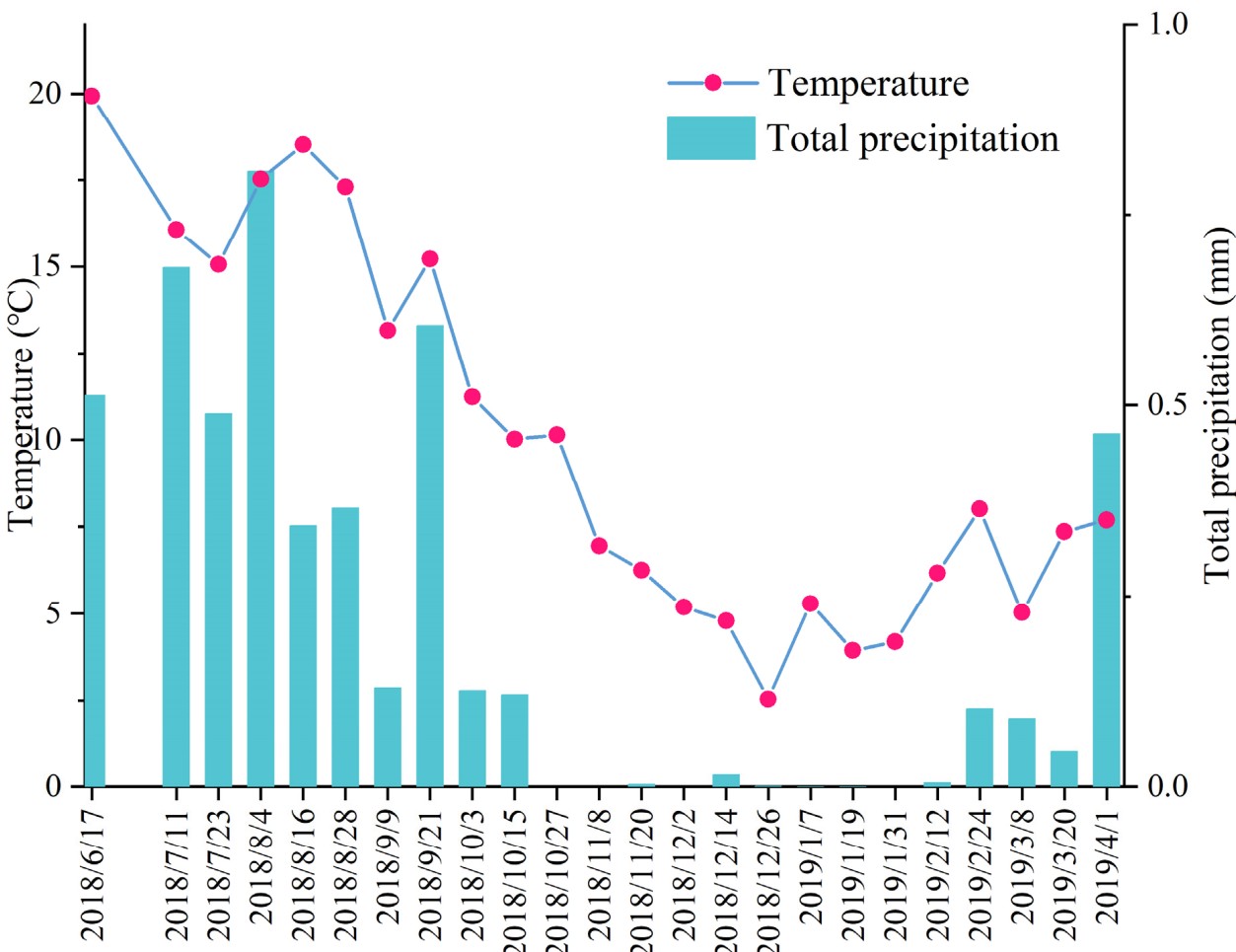

**Figure 12.** Temperature and total precipitation of ERA5 meteorological reanalysis at 11:00 UTC time.

Owing to the respective limitations of the three correction methods, such as the Linear method not being able to capture the turbulent mixing component, and the GACOS and ERA5 methods being poor at reducing the vertical stratification component, a delayed tropospheric overcorrection phenomenon occurred in the non-deformation area of the interferogram. Figures 13 and 14 show the distribution of the spatial location of the non-deformation region over the correction range in the ascending interferogram 20190119–20190212 and descending interferogram 20181204–20190121, respectively; and the phase values in the figures are the results of the original interferometric phase values minus the tropospheric corrected interferometric phase values. After correction using the three methods, a delayed overcorrection phenomenon of the troposphere appeared in most of the mountainous areas with steep topography. On both sides of the Jinsha River, the phenomenon of overcorrection after Linear method correction mostly appeared in the area with higher elevation on the back side of the mountains, along the Jinsha River, while the distribution of the overcorrection phenomenon after GACOS and ERA5 method correction was closer to the slope of the mountains along the Jinsha River. In addition, the overcorrection phenomenon was more serious in the middle part of the mountain slope in both ascending and descending track interferograms, such as the yellow and red areas in Figures 13 and 14.

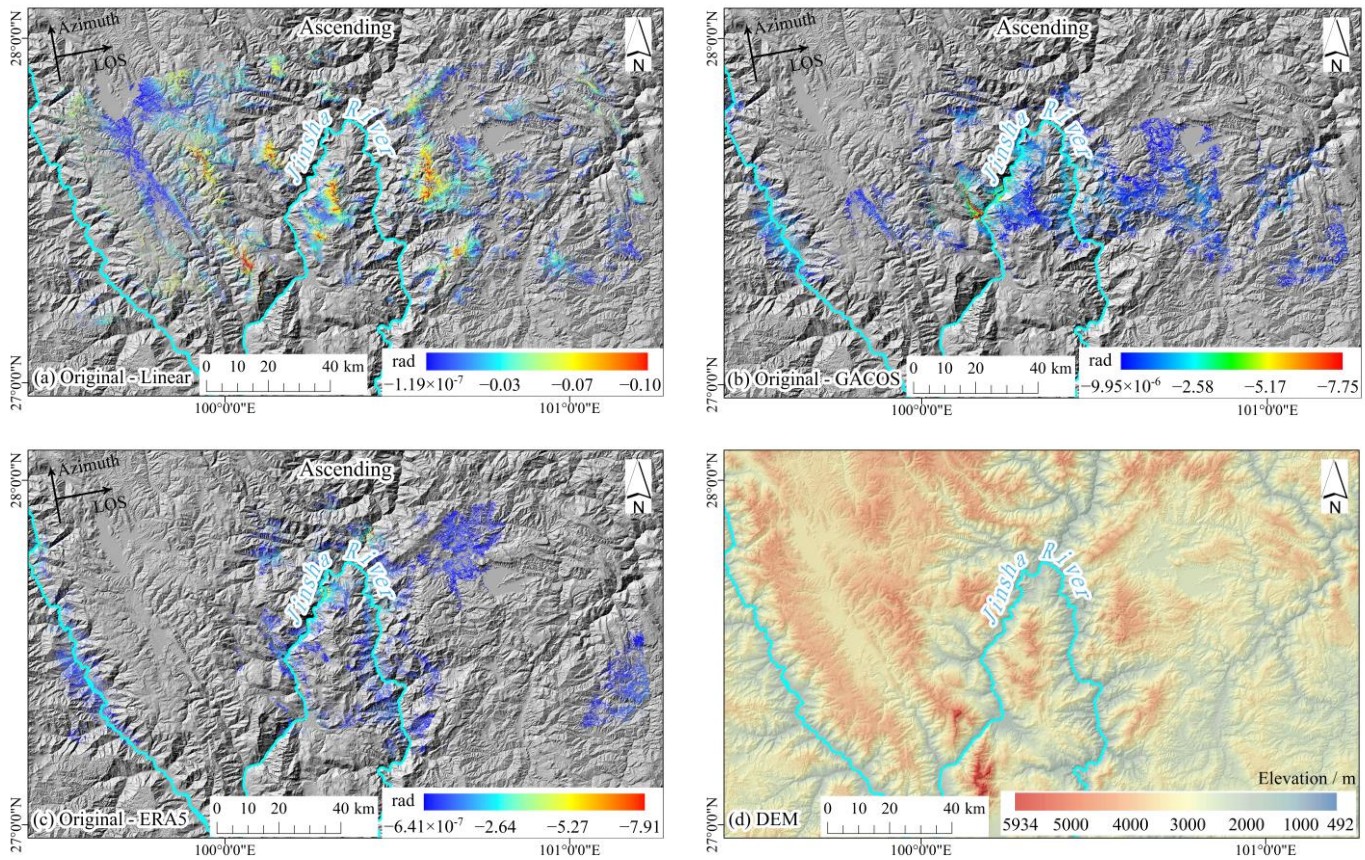

**Figure 13.** The spatial position distribution of the tropospheric delay overcorrected range in the non-deformation region of the ascending track interferogram 2019019–20190212. (**a–c**) show the spatial distribution of overcorrected range after correction using different tropospheric delay methods; (**d**) DEM of the area.

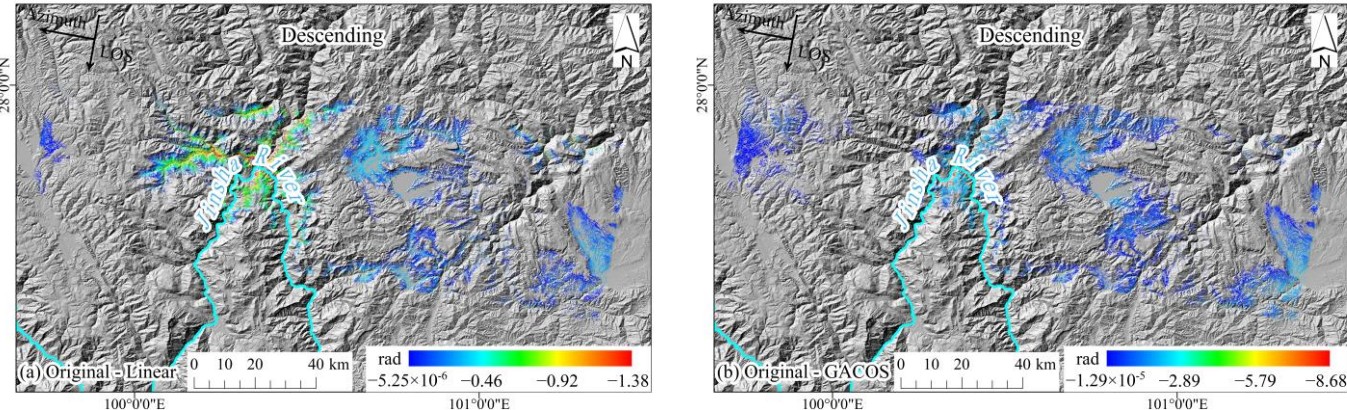

**Figure 14.** *Cont.*

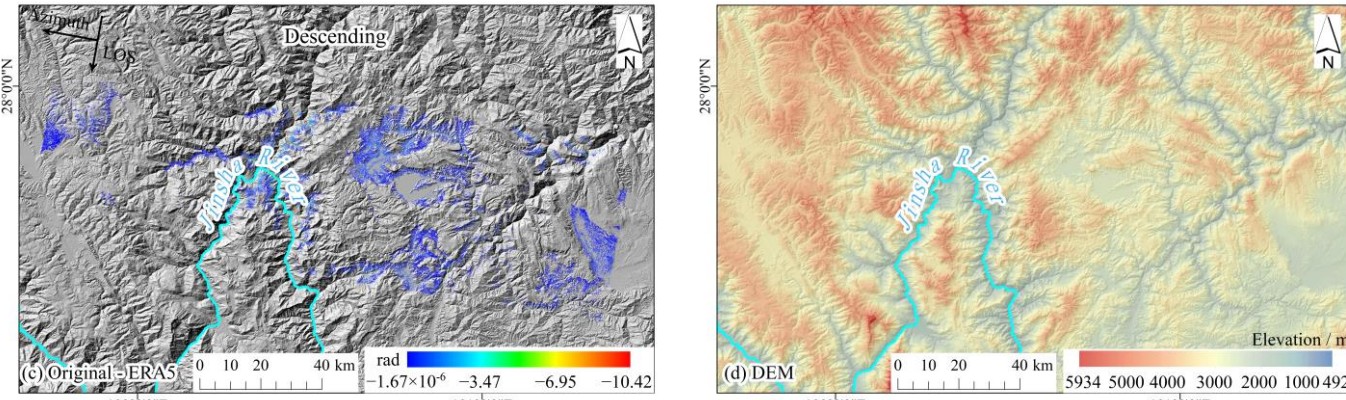

**Figure 14.** The spatial position distribution of the tropospheric delay overcorrected range in the non-deformation region of the descending tracks interferogram 20181204–20190121. (**a**–**c**) show the spatial distribution of overcorrected range after correction using different tropospheric delay methods; (**d**) DEM of the area.

We counted the distribution of elevation, slope, and slope aspect of the overcorrection phenomenon for the non-deformation area of the above two interferograms, as shown in Tables 6–8. According to the National Standard of the People's Republic of China, the slope was divided into six classes and the slope aspect was divided into nine types, and the division criteria are shown in Tables 7 and 8 [29,30]. In addition, we classified the slopes with slope aspects of northwest, north, northeast, and east as shady slopes; the slopes of southeast, south, west, and southwest as sunny slopes; and the areas without slopes as flat. Both interferograms show overcorrection in the (2000, 5000) m elevation range, and the number of overcorrected image points was the highest in the (2000, 4000) m elevation range. With the increase of altitude, the temperature and pressure in the troposphere gradually decreased, and the water vapor in the troposphere could be divided into three vertical layers, with altitudes of 0–2000 m, 2000–4000 m, and 4000–12,000 m. The water vapor content accounted for 50%, 25%, and 25% of the total atmospheric water vapor content [27,31]. The total water vapor content in the interval of elevation (2000, 4000) m was lower than that in the area below elevation 2000 m. The corresponding interferogram tropospheric delay was also lower than the tropospheric delay below 2000 m, which made it difficult to make a reasonable estimate, and this may have led to a high tendency for overcorrection of this elevation interval. Second, the area with ramp grade was more prone to overcorrection than the other slope grades. Most of the overcorrected image elements were distributed on sunny slopes in the study area, with slope aspect types of southeast, south, southwest, and west. Sunny slopes refer to slopes facing the sun, and shady slopes refer to slopes facing away from the sun, and sunlight in the northern hemisphere shines mainly from the south to the north; south-facing slopes may receive up to six-times more solar radiation than north-facing slopes, and south-facing slope areas have a more arid environment and therefore have warmer, drier, and more variable microclimates [32]. The warm and dry atmospheric environment made the difference of tropospheric activity in the sunny slope regions smaller, the value of tropospheric delay in the interferogram was smaller, and the three correction methods were less effective in correcting a smaller scale tropospheric delay, so the sunny slope was more prone to overcorrection than the shady slope, and a subsequent solution to this overcorrection phenomenon should pay attention to the effects of elevation, slope, and slope aspect.

**Table 6.** Interferogram 20190119–20190212 and 20181204–20190121 overcorrected image element elevation distribution statistics in the non-deformation region.

| Interferogram | Elevation (m) | Number of Overcorrected Image Elements (Linear) | Number of Overcorrected Image Elements (GACOS) | Number of Overcorrected Image Elements (ERA5) |
|---|---|---|---|---|
| 20190119–20190212 | [1000, 2000) | 0 | 5870 | 6963 |
| | [2000, 3000) | 8716 | 50,000 | 55,618 |
| | [3000, 4000) | 96,756 | 31,639 | 15,216 |
| | [4000, 5000) | 18,789 | 3745 | 4 |
| 20181204–20190121 | [1000, 2000] | 1147 | 1112 | 1143 |
| | [2000, 3000] | 73,970 | 62,558 | 59,025 |
| | [3000, 4000] | 10,925 | 23,226 | 7678 |
| | [4000, 5000] | 0 | 2123 | 2021 |

**Table 7.** Overcorrected image element slope grading statistics for non-deformation regions of interferograms 20190119–20190212 and 20181204–20190121.

| Interferogram | Slope Grade | Classification Criteria (°) | Number of Overcorrected Image Elements (Linear) | Number of Overcorrected Image Elements (GACOS) | Number of Overcorrected Image Elements (ERA5) |
|---|---|---|---|---|---|
| 20190119–20190212 | Flat Slope | (0, 5) | 9149 | 4437 | 2919 |
| | Gentle Slope | (5, 15) | 33,469 | 18,258 | 13,617 |
| | Ramp | (15, 25) | 42,717 | 30,603 | 25,511 |
| | Steep Slope | (25, 35) | 29,301 | 26,214 | 23,514 |
| | Rapid Slope | (35, 45) | 8158 | 9838 | 10,199 |
| | Dangerous Slope | >45 | 1467 | 1904 | 2041 |
| 20181204–20190121 | Flat Slope | (0, 5) | 7890 | 8415 | 7132 |
| | Gentle Slope | (5, 15) | 19,558 | 24,358 | 17,370 |
| | Ramp | (15, 25) | 26,581 | 29,751 | 22,689 |
| | Steep Slope | (25, 35) | 22,856 | 20,236 | 17,165 |
| | Rapid Slope | (35, 45) | 8058 | 5391 | 4851 |
| | Dangerous Slope | >45 | 1099 | 868 | 660 |

**Table 8.** Slope aspect statistics of overcorrected image elements in non-deformation regions of interferograms 20190119–20190212 and 20181204–20190121.

| Interferogram | Type | Slope Aspect | Classification Criteria (Azimuth °) | Number of Overcorrected Image Elements (Linear) | Number of Overcorrected Image Elements (GACOS) | Number of Overcorrected Image Elements (ERA5) |
|---|---|---|---|---|---|---|
| 20190119–20190212 | Plane | No slope aspect | −1 | 0 | 4 | 5 |
| | Shady Slope | Northwest | (292.5, 337.5) | 12,940 | 11,350 | 9608 |
| | | North | >337.5 or ≤22.5 | 6343 | 6940 | 6217 |
| | | Northeast | (22.5, 67.5) | 10,128 | 8857 | 7425 |
| | | East | (67.5, 112.5) | 12,976 | 9115 | 7401 |
| | Sunny Slope | Southeast | (112.5, 157.5) | 21,022 | 14,354 | 12,173 |
| | | South | (157.5, 202.5) | 26,294 | 18,055 | 15,885 |
| | | Southwest | (202.5, 247.5) | 20,174 | 12,965 | 11,035 |
| | | West | (247.5, 292.5) | 14,384 | 9614 | 8052 |
| | Total number of shady slope pixels | - | - | 42,387 | 36,262 | 30,651 |
| | Total number of sunny slope pixels | - | - | 81,874 | 54,988 | 47,145 |

**Table 8.** *Cont.*

| Interferogram | Type | Slope Aspect | Classification Criteria (Azimuth °) | Number of Overcorrected Image Elements (Linear) | Number of Overcorrected Image Elements (GACOS) | Number of Overcorrected Image Elements (ERA5) |
|---|---|---|---|---|---|---|
| 20181204–20190121 | Plane | No slope aspect | −1 | 3 | 3 | 3 |
| | Shady Slope | Northwest | (292.5, 337.5) | 6742 | 5696 | 4540 |
| | | North | >337.5 or ≤22.5 | 10,413 | 9732 | 7742 |
| | | Northeast | (22.5, 67.5) | 9357 | 9929 | 7769 |
| | | East | (67.5, 112.5) | 12,929 | 14,105 | 10,927 |
| | Sunny Slope | Southeast | (112.5, 157.5) | 16,714 | 17,746 | 14,034 |
| | | South | (157.5, 202.5) | 12,730 | 13,805 | 10,874 |
| | | Southwest | (202.5, 247.5) | 8693 | 9553 | 7452 |
| | | West | (247.5, 292.5) | 8461 | 8450 | 6526 |
| | Total number of shady slope pixels | - | - | 39,441 | 39,462 | 30,978 |
| | Total number of sunny slope pixels | - | - | 46,598 | 49,554 | 38,886 |

The results show that the three correction methods could alleviate the tropospheric delay to varying degrees. The GNSS monitoring results verified that the methods improved the InSAR deformation monitoring accuracy at some stations. However, because of their small coverage area, they only represent the tropospheric delay correction situation in a small-scale space and could not reflect the overall situation of the study area. Therefore, based on the above results of the $STD_1$, semi-variance function, elevation correlation evaluation, and GNSS verification evaluation, we showed that in the low-latitude alpine canyon region with steep terrain of the study area, the InSAR tropospheric delay was dominated by the vertical stratification component, and the Linear correction method greatly improved the deformation monitoring accuracy, followed by the GACOS method. Meanwhile, the ERA5 method had a poor stability but a better correction effect than the above two methods; however, it could also improve the accuracy of deformation monitoring. The Linear method of the empirical model correction could only estimate the stratified component of the tropospheric delay. To estimate the turbulent mixed component, estimation using a correction method of an atmospheric numerical model can be supplemented [33], and the advantages of the two types of correction methods could be combined to further alleviate the tropospheric delay.

## 5. Conclusions

In this study, $STD_1$, semi-variance function, elevation correlation, and GNSS deformation monitoring results were used to evaluate the tropospheric delay correction effect of the Linear correction, GACOS, and ERA5 methods in a low-latitude plateau canyon region. After correction using the Linear method, the Aver of the interferogram was decreased by –20.98%, the mean value of the threshold value of sill was decreased by –41%, and the accuracy of the InSAR deformation points near the GNSS sites increased by 58%. The three methods effectively alleviated tropospheric delays in InSAR interferometry. The Linear method improved the accuracy of InSAR deformation monitoring to the greatest extent and provided more accurate reference data for surface motion research.

Based on the evaluation results of the above criteria, the following conclusions were drawn: (1) in low-latitude alpine canyon regions, where the tropospheric delay is dominated by the vertical stratified component, the Linear correction method is the most suitable, followed by the GACOS method, whereas the ERA5 method has poor correction stability; (2) for long time series InSAR deformation monitoring, if the number of images allows, SAR images during winter imaging can largely avoid the influence of tropospheric disturbances in interferometry.

**Author Contributions:** Y.Z. and X.Z. conceived the manuscript; Y.Z. interpreted the results and drafted the manuscript: X.Z., J.B., Y.L., S.G. and Q.Y. checked and revised the manuscript; Y.Z., X.Z., J.B., Y.L. and S.G. conducted the experiments and obtained the results; Y.Z., X.Z., Y.L., S.G. and Q.Y. contributed to the discussion of the results; X.Z. provided funding support. All authors have read and agreed to the published version of the manuscript.

**Funding:** This research was funded by the National Natural Science Foundation of China (Grant No. 42161067).

**Data Availability Statement:** Not applicable.

**Acknowledgments:** The authors wish to thank the European Space Agency for providing the Copernicus Sentinel-1A SAR data, NASA and NIMA for providing the SRTM-1 DEM data, and the European Space Agency for providing the POD data. GACOS products were provided by Newcastle University at http://www.gacos.net/ (accessed on 1 September 2022). ERA-5 data were provided by the European Centre for Medium-Range Weather Forecasts (ECMWF) at https://cds.climate.copernicus.eu/cdsapp#!/search?type=dataset (accessed on 1 September 2022). InSAR processing was performed using ISCE, StaMPS, and TRAIN software. All authors agree and thank the anonymous commenters for their in-depth comments and helpful suggestions, which helped to greatly improve this article.

**Conflicts of Interest:** The authors declare no conflict of interest.

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
