# Peer review of "Evaluation of InSAR Tropospheric Delay Correction Methods in a Low-Latitude Alpine Canyon Region"

_remotesensing, doi:10.3390/rs15040990_

Round 1

Reviewer 1 Report

The article was prepared correctly and carefully. This is testing known and available methods of reducing the impact of the troposphere, so it is not an original and innovative development. Nevertheless, it is an interesting case study worth publishing. I suggest improving the quality of charts and drawings in terms of their readability and appearance.

Author Response

Dear Reviewer,

       Many thanks to the reviewers for their comments. We have carefully studied the comments and made revisions, and hope to receive your approval. The details of the revisions are shown in the PDF attachment.

Reviewer 2 Report

see attached

Author Response

(The authors gave the same response as above.)

Reviewer 3 Report

REVISION MANUSCRIPT Remote Sensing- 2168932: Evaluation of InSAR tropospheric delay correction methods in low latitude alpine canyon region

 General comments:

The authors proposed a correction method to improve the accuracy of InSAR deformation monitoring for a high mountain region in China. In order to accomplish that, the authors also considered Sentinel-1A images of ascending and descending tracks, some meteorological reanalysis data (ERA5) methods and GNSS. Therefore, I found the research very interesting. However, minor revisions need to be applied in order to consider it for publication.

Specific comments:

Line 214, Resolution of Figure 1 (a) and (b) must be improved.

Line 244, Resolution of Figure 2 should be improved.

Line 259, Resolution of Figure 3 (a) and (b) must be improved.

Line 344, Resolution of Figure 4 (a) and (b) must be improved.

Line 347, Resolution of Figure 5 (a) and (b) must be improved.

Line 369, Resolution of Figure 6 (a) - (f) must be improved.

Line 431, Resolution of Figure 7 (a) and (b) must be improved.

Line 467, Resolution of Figure 8 (a) and (b) must be improved.

Line 486, Resolution of Figure 6 (a) - (h) must be improved.

Line 490-493, Could you please provide more specifications for the 12 GNSS Stations used in terms of precision and/or accuracy?

Line 495-496, “Because the GNSS deformation monitoring date was not completely consistent with the InSAR image acquisition time”…Why was that? How this impact/affect your solution?

Line 496-497, “it was compared with the deformation value at the nearest moment of the InSAR reference image acquisition date,…” Is this valid? Prove it or provide a reference!

Line 503-504, ??, ??, and ?? represent the north, east, and vertical deformation values of the GNSS stations, respectively.” How accurate are these values?

Line, 508-510: “The difference between the InSAR deformation value of the descending tracks and the GNSS monitored deformation value was several times.” …difference: several times!! This sentence is confused needs to be rewritten.

Line 522, Resolution of Figure 10 (a) - (d) must be improved.

Line 579, Resolution of Figure 11 (a) and (b) must be improved.

Line 583, Resolution of Figure 13 should be improved. Also, it must read: Figure 12 instead of Figure 13.

Author Response

(The authors gave the same response as above.)
